# Distinct adaptation and epidemiological success of different genotypes within *Salmonella enterica* serovar Dublin

**Cheryll M Sia[1,2]\*, Rebecca L Ambrose[3], Mary Valcanis[2], Patiyan Andersson[2], Susan A Ballard[2], Benjamin P Howden[1,2], Deborah A Williamson[4†], Jaclyn S Pearson[3,4,5]\*†, Danielle J Ingle[1]\*†**

[1]Department of Microbiology and Immunology, The University of Melbourne at The Peter Doherty Institute for Infection and Immunity, Melbourne, Australia; [2]Microbiological Diagnostic Unit Public Health Laboratory, Department of Microbiology and Immunology, The University of Melbourne at The Peter Doherty Institute for Infection and Immunity, Melbourne, Australia; [3]Centre for Innate Immunity and Infectious Diseases, Hudson Institute of Medical Research, Clayton, Australia; [4]School of Medicine, University of St Andrews, St Andrews, United Kingdom; [5]Department of Microbiology, Monash University, Clayton, Australia

**\*For correspondence:**
cmsia@unimelb.edu.au (CMS);
jp287@st-andrews.ac.uk (JSP);
danielle.ingle@unimelb.edu.au (DJI)

†These authors contributed equally to this work

## eLife Assessment

This study provides the most comprehensive analysis of *Salmonella* Dublin to date, uncovering distinct genotypic adaptations, antimicrobial resistance patterns, and virulence strategies that influence epidemiological success. The revised manuscript is very **valuable**, rigorous and **compelling**.

**Abstract** *Salmonella* Dublin is a host-adapted, invasive nontyphoidal *Salmonella* (iNTS) serovar that causes bloodstream infections in humans and demonstrates increasing prevalence of antimicrobial resistance (AMR). Using a global dataset of 1303 genomes, coupled with in vitro assays, we examined the evolutionary, resistance, and virulence characteristics of *S*. Dublin. Our analysis revealed strong geographical associations between AMR profiles and plasmid types, with highly resistant isolates confined predominantly to North America, linked to IncC plasmids co-encoding AMR and heavy metal resistance. By contrast, Australian isolates were largely antimicrobial-susceptible, reflecting differing AMR pressures. We identified two phylogenetically distinct Australian lineages, ST10 and ST74, with a small number of ST10 isolates harbouring a novel hybrid plasmid encoding both AMR and mercuric resistance. Whereas the ST10 lineage remains globally dominant, the ST74 lineage was less prevalent. ST74 exhibited unique genomic features including a larger pan genome compared to ST10 and the absence of key virulence loci, including *Salmonella* pathogenicity island (SPI)-19 which encodes a type VI secretion system (T6SS). Despite these genomic differences, the ST74 lineage displayed enhanced intracellular replication in human macrophages and induced less pro-inflammatory responses compared with ST10, suggesting alternative virulence strategies that may support systemic dissemination of ST74. The Vi antigen was absent in all ST10 and ST74 genomes, highlighting challenges for serotyping and vaccine development, and has implications for current diagnostic and control strategies for *S*. Dublin infections. Collectively, this study represents the most comprehensive investigation of *S*. Dublin to date and, importantly, has revealed distinct adaptations of two genotypes within the same serovar, leading to different epidemiological success. The regional emergence and evolution of distinct *S*. Dublin lineages highlight the need to understand the divergence of intra-serovar virulence mechanisms

which may impact the development of effective control measures against this important global pathogen.

## Introduction

Invasive nontyphoidal *Salmonella* (iNTS) infections are a major cause of global morbidity and mortality (*Stanaway et al., 2019*). Of the >2500 NTS serovars, iNTS infections are associated with a relatively limited number of serovars. These include *Salmonella enterica* serovar Dublin (*S.* Dublin), *S.* Panama, *S.* Virchow, *S.* Choleraesuis, and *S.* Typhimurium ST313 (*Parisi et al., 2019*; *Mughini-Gras et al., 2020*; *Kingsley et al., 2009*). *S.* Dublin, a bovine-adapted pathogen, is one of the leading iNTS serovars both in Australia and globally (*Cheng et al., 2019a*; *Williamson et al., 2018*; *Mangat et al., 2019*) and is associated with bacteraemia in a high proportion of infections (*Parisi et al., 2019*; *Cheng et al., 2019a*; *Williamson et al., 2018*; *Mangat et al., 2019*). Symptomatic cattle may develop septicaemia and transplacental transmission (*Mangat et al., 2019*; *Reichel et al., 2018*), while asymptomatic infections in cattle enable persistence within the herd population, facilitating disease spread (*Mangat et al., 2019*; *Kirchner et al., 2012*). *S.* Dublin causes zoonotic infections predominately through the consumption of contaminated dairy and meat produce (*McDonough et al., 1999*; *Davidson et al., 2018*; *Fenske et al., 2019*), and as such is considered a 'One Health' pathogen (*McDonough et al., 1999*; *Davidson et al., 2018*; *Fenske et al., 2019*).

Antimicrobial resistance (AMR) in NTS has increased over the past several decades, leading to the reduction of therapeutic options (*Williamson et al., 2018*; *Sia et al., 2021*). Although *S.* Dublin isolates are largely susceptible to therapeutics, AMR has become increasingly prevalent in *S.* Dublin (*Mangat et al., 2019*; *Fenske et al., 2019*; *Paudyal et al., 2019*; *Harvey et al., 2024*; *Hsu et al., 2019*). For example, a multidrug-resistant (MDR; defined as resistance to ≥3 antimicrobial classes [*Magiorakos et al., 2012*]) lineage has recently been described in North America, characterised by varying profiles of AMR to ampicillin, streptomycin, chloramphenicol, sulphonamides, tetracyclines, and third-generation cephalosporins (3GC). These MDR *S.* Dublin isolates all type as sequence type 10 (ST10), and the AMR determinants have been demonstrated to be carried on an IncC plasmid that has recombined with a virulence plasmid encoding the *spvRABCD* operon (*Mangat et al., 2019*; *Fenske et al., 2019*; *Hsu et al., 2019*; *Mangat et al., 2017*). This has resulted in hybrid virulence and AMR plasmids circulating in North America, including a 329 kb megaplasmid with IncX1, IncFIA, IncFIB, and IncFII replicons (isolate CVM22429, NCBI accession CP032397.1) (*Fenske et al., 2019*; *Hsu et al., 2019*) and a smaller hybrid plasmid 172,265 bases in size with an IncX1 replicon (isolate N13-01125, NCBI accession KX815983.1) (*Mangat et al., 2017*).

*S.* Dublin is closely related to *S.* Enteritidis and *S.* Gallinarum (*Langridge et al., 2015*), with differences in accessory genome content reflective of adaptation to different ecological niches (*Langridge et al., 2015*). The majority of the sequence types (STs) reported for *S.* Dublin are ST10 or single-locus variants of this ST, although there is a second lineage, ST74, that is distinct from the main *S.* Dublin population (*Achtman et al., 2012*). This uncommon *S.* Dublin lineage, ST74, may be variably serotyped as *S.* Dublin or *S.* Enteritidis dependent upon the strains used to generate typing sera in different laboratories (*Achtman et al., 2012*), which may have also contributed to it not being identified. The antigenic formula of *S.* Dublin (1,9,12:g,p:-) is highly similar to *S.* Enteritidis (1,9,12:g,m:-), with three non-synonymous point mutations in *fliC* (Ala220Val, Thr315Ilc, and Thr318Ala) differentiating the two serovars. Although the Vi capsular antigen (encoded by the *viaB* locus) is part of the antigenic definition for *S.* Dublin, it has been shown that the Vi antigen is absent in many *S.* Dublin isolates (*Achtman et al., 2012*; *Selander et al., 1992*). Other putative virulence determinants in *S.* Dublin include *Salmonella* pathogenicity islands (SPIs) 6 and 19, which both encode type VI secretion system (T6SS) machineries (*Pezoa et al., 2014*; *Blondel et al., 2009*). The presence of two T6SSs in *S.* Dublin contrasts with other NTS serovars, which encode a single T6SS (*Pezoa et al., 2014*; *Blondel et al., 2009*). T6SSs are molecular nanomachines that deliver effector proteins into eukaryotic and/or prokaryotic cells to promote pathogen survival within multispecies and complex immune environments such as the gut. Recent studies have demonstrated a role for SPI-6 and SPI-19 in interbacterial competition by specific *S.* Dublin strains (*Blondel et al., 2023*; *Amaya et al., 2022*); however, their effect on host immune processes remains largely unknown.

**eLife digest** *Salmonella* bacteria often live harmlessly in animals but can make people sick. For example, milk or meat from cows infected with a type called *Salmonella* Dublin can cause serious blood infections. Over time, many *Salmonella* strains have become harder to treat because they carry genes that let them survive antibiotic treatment. These survival genes often sit on small DNA circles called plasmids, which can move between bacteria.

By studying the DNA from *Salmonella* Dublin samples from around the world, scientists can learn how different groups of this bacterium have evolved to resist drugs and cause infections in humans. Understanding these changes may help explain why some types of *Salmonella* Dublin spread more easily to humans and may point to better ways to detect and control infections.

Sia et al. identified two main lineages of *Salmonella* Dublin: the widespread ST10 group and a less common ST74 group. The experiments compared DNA from more than 1300 strains of *Salmonella* Dublin from 13 countries on five continents. The ST10 strains from North America often had genes that made them resistant to antibiotics and heavy metals. Resistance to heavy metals may make it easier for bacteria to develop resistance to drugs. Most Australian strains of ST10 remained drug sensitive. ST74 strains lack a key secretion system gene set but have more genes overall than ST10. In laboratory tests using human immune cells, ST74 bacteria multiplied up to eight times more over 24 hours and triggered weaker immune responses in the cells than ST10. Both groups of bacteria lacked the Vi capsule gene, which some vaccines against bacterial infections have targeted.

The experiments may also help explain why some strains of the bacteria cause more disease and point to new ways to prevent or treat infections. Knowing that ST74 does not trigger a strong immune response may help scientists develop new therapies to strengthen the immune response and prevent the bacteria from multiplying and spreading. Vaccines targeting the Vi capsule may not be effective against *Salmonella* Dublin, because most strains do not have it. Instead, scientists may want to use the data to look for alternate targets. More research is needed to support the study's findings.

We previously undertook a genomic investigation of another emerging NTS lineage in Australia, *S. enterica* serovar 4,[5],12:i:-, and identified several bacterial traits that may have contributed to the successful spread of this lineage (*Vilela et al., 2020*; *Hammarlöf et al., 2018*; *Herrero-Fresno et al., 2018*; *Mohammed and Cormican, 2016*; *De Sousa Violante et al., 2022*). In particular, we hypothesised that flagellin deletion and MDR status may have contributed to host immune evasion and positive selection, respectively, in *S. enterica* serovar 4,[5],12:i:- (*Ingle et al., 2021*). Here, we sought to understand the contribution of virulence and resistance to the recent emergence of MDR *S.* Dublin. We interrogated a global dataset of 1303 isolates collected from 13 countries on five continents and identified distinct geographical lineages associated with specific AMR and virulence profiles, including a lineage unique with increased capability for intracellular survival and immune evasion in human macrophages.

## Results

### Distinct *S.* Dublin populations circulate globally

A total of 1303 *S.* Dublin genomes were included in our analysis, spanning four decades (*Figure 1A*, *Supplementary file 1A and B*). This dataset included 53 genomes from humans in Australia, and 1250 genomes from Africa, Europe, North and South America collected from humans (n=462), animals (n=551), food (n=223), and other sources that were not stated (n=67) (*Figure 1B*, *Supplementary file 1A and B*). In total, 1235 genomes (99.4%) were ST10 and 58 were a single locus variant (SLV) of ST10, with two double locus variant genomes (e.g. ST4100) restricted to isolates from South America. The exceptions were ST74 and SLV ST1545 that were only detected in Australia (*Figure 1C–E*, *Supplementary file 1C*). All isolates, including the ST74 lineage, were serotyped in silico as *S.* Dublin. However, the ST74 lineage and a single ST10 isolate had codon profiles in *fliC* of Gly60, Leu138, Ala220, Thr315, and Ala318, in contrast to the remaining ST10 and SLV, which both had a profile of Gly60, Leu138, Val220, Ile315, and Ala318 (the previously described 'Du2' and 'Du1' electrophoretic profiles [*Selander et al., 1992*], respectively) (*Supplementary file 1D*).

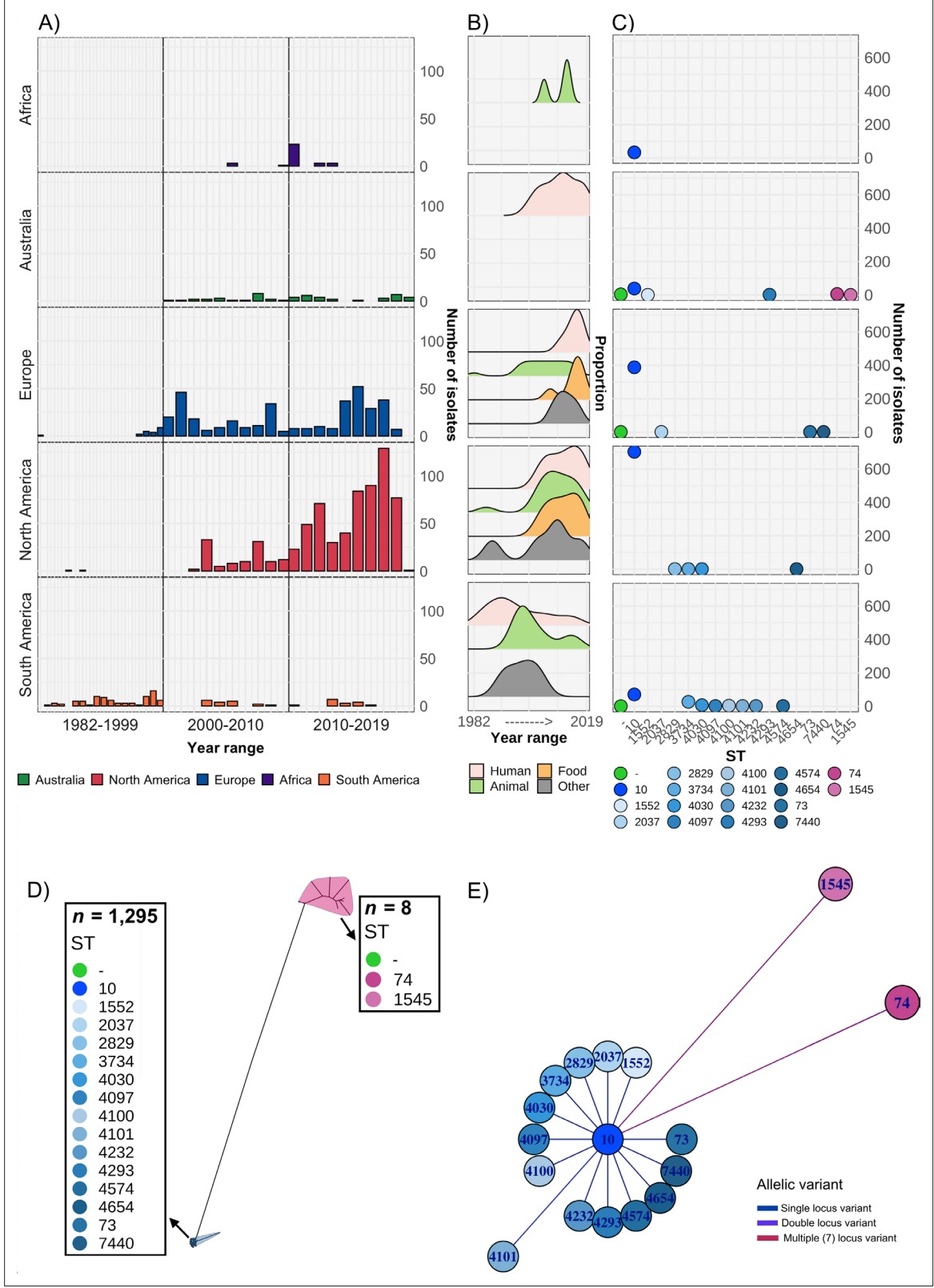

**Figure 1.** Overview of *S.* Dublin genomes in the study. Geographical distribution and clonal relationship of 1303 *S.* Dublin genomes in the study. (**A**) Number of *S.* Dublin genomes according to geographical region and time interval of collection. (**B**) Distribution of human, animal, and food *S.* Dublin genomes collected over time. (**C**) Distribution of *S.* Dublin sequence types (STs) from each region. Each coloured circle correlates to an ST. Circles in green indicate no ST was identified. (**D**) Unrooted phylogenetic tree of 1303 *S.* Dublin genomes. Two main lineages of ST74 (n=8) and

*Figure 1 continued on next page*

*Figure 1 continued*

sequence type 10 (ST10) (n=1295) shown in blue and pink, respectively. (**E**) Relatedness of known STs to ST10. The nodes are labelled with the ST, and the branch lengths are indicative of the number of dissimilar locus variants to ST10.

The online version of this article includes the following figure supplement(s) for figure 1:

**Figure supplement 1.** Phylogeny of 1303 *S.* Dublin genomes.

A maximum likelihood (ML) phylogeny was inferred from the 1303 genomes, revealing two major lineages of *S.* Dublin, ST74 and ST10 (*Figure 1D*, *Figure 1—figure supplement 1*). Within the ST10 lineage, five clades were identified using Bayesian analysis of population structure (BAPS), inferred from the alignment of 2957 core single nucleotide polymorphisms (SNPs). Four clades were broadly associated with geographical region (*Figure 1—figure supplement 1*, *Supplementary file 1A*). The Australian ST10 isolates (n=39) clustered in clade 1, while isolates from North America were associated with clade 5 (*Figure 1—figure supplement 1*). The BAPS clade 2 were the isolates that didn't have sufficient evidence to be split into further clades, hence are spread across the tree spanning a range of geographical regions.

## Emergence of *S.* Dublin lineages in the 20th century

To investigate the emergence of *S.* Dublin clades, Bayesian phylodynamic analysis was undertaken on a representative subset of 660 genomes, selected for phylogenetic, geographical, and AMR diversity (see Methods). We inferred a maximum clade credibility (MCC) phylogeny from an alignment of 3971 SNPs using the highest supported model of a relaxed lognormal molecular clock and coalescent exponential tree prior. The estimated substitution rate was $1.05 \times 10^{-7}$ (95% highest posterior density [HPD]: $1.16 \times 10^{-7}$–$9.41 \times 10^{-8}$), slightly higher than previous estimated mean substitution rates for host-restricted *Salmonella* serovars (*S.* Typhi and *S.* Paratyphi A) but lower than rates reported for host-generalist NTS serovars *S.* Kentucky and *S.* Agona (*Duchêne et al., 2016*).

The topology of the MCC tree was consistent with the ML tree, with geographically associated sub-clades observed in the MCC tree (*Figure 2*, interactive tree is available in Microreact under https://microreact.org/project/sdublinpaper). The most recent common ancestor (MRCA) for the *S.* Dublin's global population was inferred to be 1901 (HPD 1878–1924), with the MRCA of the different clades and sub-clades largely falling between 1950 and 2000. For example, the MRCA of clade 5 MDR isolates from North America was estimated to be ~1964 (HPD 1952–1974) (*Figure 2*). The MRCA of the Australian sub-clade was estimated to be ~1967 (HPD 1953–1980), with the Australian and North American isolates inferred to have an MRCA~1931 (HPD 1914–1947). Moreover, three clades (1, 3, and 4) had European sub-clades associated with Sweden, the UK, and Denmark, respectively, that were all largely susceptible to antimicrobials. Some regions, such as Europe and South America, were associated with several sub-clades of *S.* Dublin, potentially suggesting multiple importations and expansions, while other regions, such as Australia, were comprised of a single sub-clade within clade 1.

## Resistance determinants are associated with geographical regions

Different plasmid types co-occurred with different AMR and heavy metal resistance (HMR) determinants in the *S.* Dublin population, and these were also associated with differential geographical distribution (*Figure 2*, *Figure 3—figure supplement 1*, *Figure 3—figure supplement 2*, *Figure 3—figure supplement 3*, *Supplementary file 1A*). Overall, 51.5% (667/1,295) of *S.* Dublin genomes had at least one AMR determinant, with 632/667 (94.8%) isolates classified as MDR. MDR *S.* Dublin was predominantly identified from North American isolates, although some AMR determinants were detected in *S.* Dublin from other geographical regions (*Figure 3*, *Figure 3—figure supplement 1*, *Figure 3—figure supplement 2*). AMR *S.* Dublin collected from animals was common (n=305/667; 45.7%). AMR in *S.* Dublin from food sources and humans was similar in proportion (n=174/667; 26.1% and n=173/667; 25.9% respectively). Although these profiles are likely impacted by the sampling frame of different studies, the source of collection was not considered as a selection criterion for this study, and as such, may broadly represent the AMR profiles of *S.* Dublin from different sources (*Figure 3A and B*, *Figure 1—figure supplement 1*).

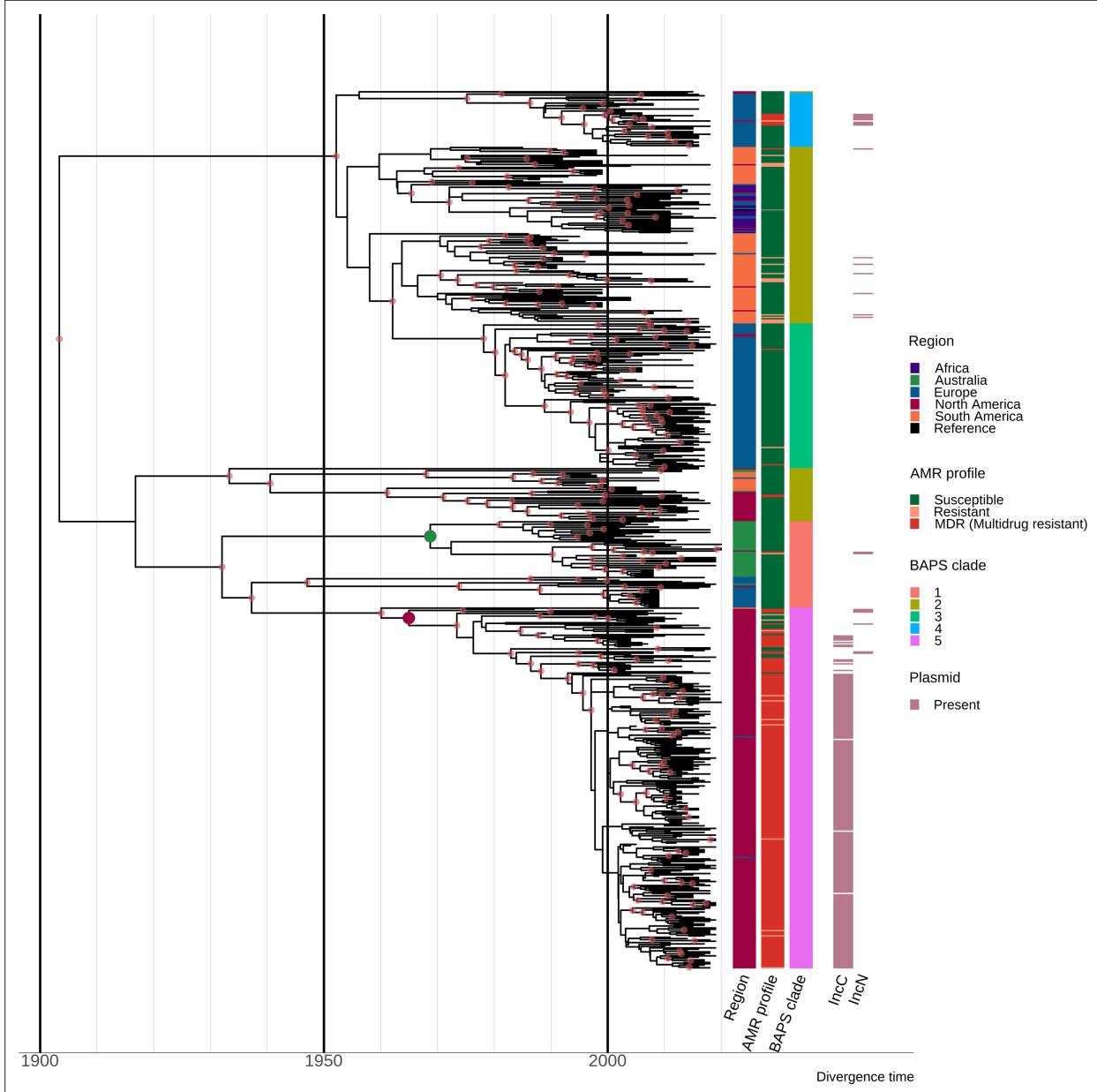

**Figure 2.** Estimation of divergence time. Maximum clade credibility (MCC) tree of the 667 *S.* Dublin sequence type 10 (ST10) genomes. The time in years is shown on the x-axis. From left to right, the coloured heatmaps refer to the region of origin, the susceptibility profile, the Bayesian analysis of population structure (BAPS) clade, and the presence of IncC and IncN plasmid types. The green dot denotes the most recent common ancestor (MRCA) of the Australian subclade, while the maroon dot represents the MRCA of clade 5 multidrug-resistant (MDR) isolates from North America. Internal nodes coloured chestnut red have a posterior probability ≥0.95.

The online version of this article includes the following figure supplement(s) for figure 2:

**Figure supplement 1.** Comparison of plasmids from antimicrobial-resistant AUSMDU00035676 and AUSMDU00016678.

**Figure supplement 2.** Plasmid assembly of a virulence plasmid identified in an *S.* Dublin sequence type 10 (ST10) isolate mapped to reference IncX1/IncFII(S) virulence plasmids.

The most prevalent AMR determinants were *strA-B* (n=620), *floR* (n=556), *tet*(A) (n=608), and *sul2* (n=596) (*Figure 3—figure supplement 1*). AMR mechanisms to ciprofloxacin (triple point mutations in quinolone resistance determining regions *gyrA*[83:S-F], *gyrA*[87:D-N], *parC*[80:S-I]), macrolides (*mph*(A)), and colistin (*mcr-9.1*) were each detected in single genomes from *S.* Dublin clade 5 at a low frequency of 0.08% (n=1/1303) in the dataset. The only 3GC resistance mechanism detected

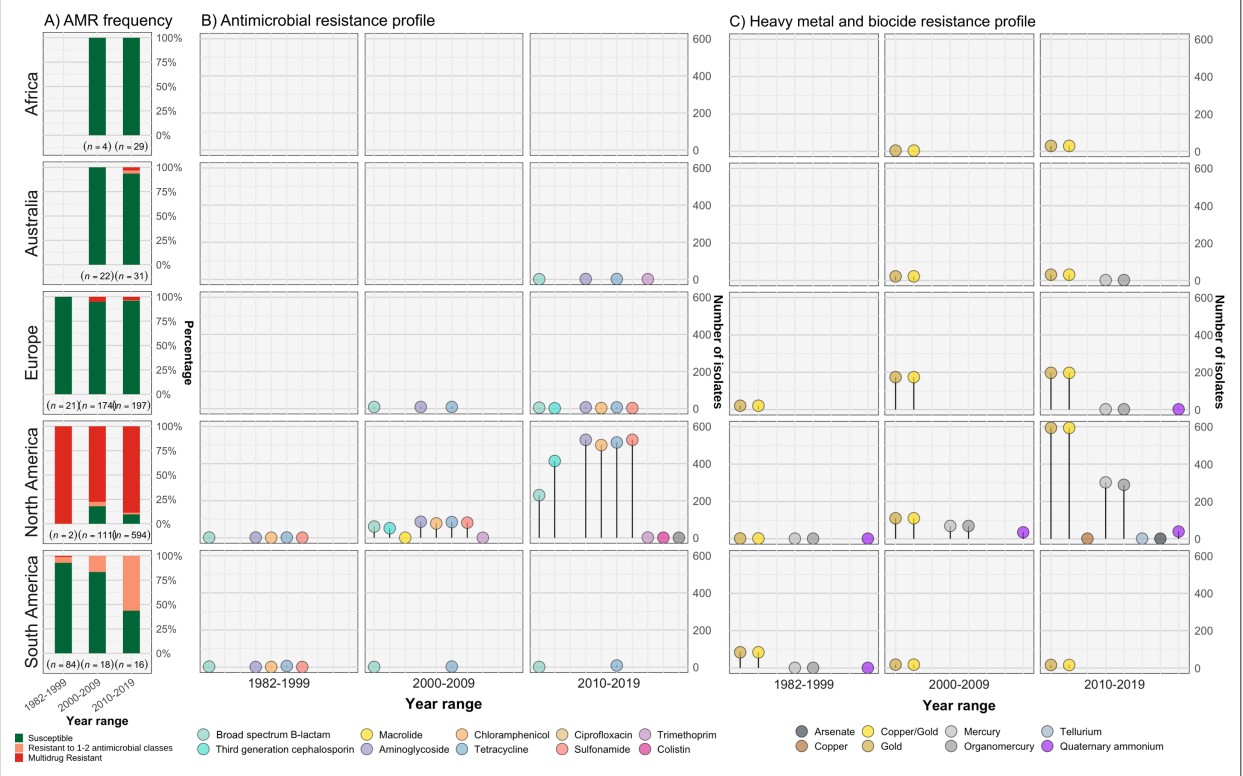

**Figure 3.** Resistance profiles of *S*. Dublin genomes. Observed resistance profiles of *S*. Dublin genomes spanning three time intervals for each region. (**A**) Frequency of susceptible and resistant *S*. Dublin genomes from each time period. (**B**) Antimicrobial resistance profiles observed spanning three time periods. The colour of the circle corresponds to an antimicrobial class, while the height is relative to the number of genomes. (**C**) Heavy metal and biocide resistance profiles observed. The colour of the circle corresponds to its respective heavy metal/biocide class, while the height is relative to the number of genomes.

The online version of this article includes the following figure supplement(s) for figure 3:

**Figure supplement 1.** Resistant genotypic profiles of *S*. Dublin genomes.

**Figure supplement 2.** Distribution of resistance determinants in 1303 *S*. Dublin genomes.

**Figure supplement 3.** Co-occurrence networks of IncN and IncC resistant profiles.

was *bla*CMY-2, present in 467/667 (70%) of AMR *S*. Dublin genomes. Although *strA-B*, *tet*(A), and *bla*TEM-1 were observed in multiple clades, *floR*, *sul2*, and *bla*CMY-2 were restricted to North American isolates that fell in clade 5 (*Figure 3—figure supplement 1*, *Figure 3—figure supplement 2*). This MDR profile, in addition to *strA-strB* and *tet*(A), was mediated by IncC plasmids in 593 genomes (n=509 from the USA, n=82 from Canada, and n=2 from the UK) (*Figure 3—figure supplement 2*, *Figure 3—figure supplement 3C*). Nearly all clade 3 isolates linked to the UK had no AMR determinants, except for four isolates collected between 2010 and 2015 (*Figure 1—figure supplement 1*, *Figure 3*, *Figure 3—figure supplement 1*, *Figure 3—figure supplement 2*, *Supplementary file 1A*).

In addition to AMR determinants, we characterised biocide and HMR profiles of the *S*. Dublin isolates. Gold resistance determinants *golS* and *golT* were ubiquitous in *S*. Dublin (n=1302/1303; 99.9%) (*Figure 3C*, *Figure 3—figure supplement 1*, *Figure 3—figure supplement 2*). Consistent with previous findings, biocide and HMR determinants, including *qacEdelta1* and mercury resistance genes (*merABDEPRT*), commonly co-occurred with AMR determinants. Notably, the *mer* operon was frequently associated with the IncC MDR plasmid (90.1%; n=534/593) (*Figure 3—figure supplement 2*, *Supplementary file 1A*).

## Characterisation of S. Dublin plasmids and identification of a novel hybrid plasmid

Both IncX1 (n=1269/1303; 97.4%) and IncFII(S) (n=1213/1303; 93.1%) plasmid replicons were found at high frequency within the dataset, in line with previous findings (*Fenske et al., 2019*). Three plasmid replicons, namely IncN, IncX1, and IncFII(S), and an MOBF/MOBP relaxase were detected in one of the Australian *S.* Dublin ST10 genomes from clade 1 (AUSMDU00035676). Comparative analysis revealed the AUSMDU00035676 plasmid lacked the *spv* operon and had low homology to reference IncN and IncX1/IncFII(S) plasmids previously associated with *Salmonella* (*Figure 2—figure supplement 1*, *Supplementary file 1E*). The loss of the *spv* operon may be associated with a less invasive phenotype (*Silva et al., 2017*). This finding suggests the detection of a potentially novel hybrid plasmid with IncN/IncX1/IncFII(S) markers that encodes AMR and HMR determinants flanked by IS26 elements, which may aid gene mobilisation (*Harmer and Hall, 2020*). In contrast, the AUSMDU00056868 plasmid had IncX1/IncFII(S) plasmid replicon markers and was found to not encode any mechanisms for resistance. This plasmid was >99% homologous to known virulence reference *S.* Dublin plasmids pOU1115 and pSDU2_USMARC_69807 and encoded the *spv* operon (*Supplementary file 1E*, *Figure 2—figure supplement 2*). This suggests that the *S.* Dublin virulence plasmid is conserved, while the *spv* operon that has been shown to enhance disease severity (*Paudyal et al., 2019*; *Hsu et al., 2019*; *Mohammed et al., 2017*; *Hong et al., 2008*; *Chu et al., 2008*) is maintained on the plasmid. Complete annotations of both AUSMDU00035676 and AUSMDU00056868 can be found in *Supplementary file 1F and G*, respectively.

## Diversity in virulence determinants across *S.* Dublin lineages

To investigate the potential for invasiveness within *S.* Dublin, we applied a previously described bioinformatic tool to our dataset of 1303 *S.* Dublin genomes (*Wheeler et al., 2018*). In brief, this tool infers the likelihood of invasiveness by assigning an index score between 0 and 1 that reflects the level of genome degradation, whereby a higher score suggests genomic features that may be more associated with a narrower host range and greater likelihood of invasiveness (*Wheeler et al., 2018*). We performed initial validation work using 36 complete publicly available reference genomes, covering known invasive and non-invasive phenotypes (*Supplementary file 1H*). Using this validation set, invasive and largely host-restricted serovars, including *S.* Choleraesuis, *S.* Paratyphi, and *S.* Typhi, scored >0.5, suggestive of an invasive genomic profile (*Figure 4A*, *Supplementary file 1H*). Conversely, non-invasive NTS serovars with broader ecological niches, including *S.* Anatum, ST19 *S.* Typhimurium, and *S.* Derby, scored <0.5 (*Figure 4A*, *Supplementary file 1H*). When applied to our *S.* Dublin dataset, only 11 genomes (n=11/1303; 0.84%) had an index score <0.5. These included three ST10 genomes and all eight ST74 genomes (*Figure 4B*, *Supplementary file 1I*). Based on descriptive analysis, no apparent relationship was observed between index scores and region or source attribution (*Figure 4B*).

Putative virulence determinants identified in *S.* Dublin genomes included SPIs, the *spv* operon (*Hsu et al., 2019*), prophages, and other individual genes (*Supplementary file 1I*). Some putative virulence factors, such as *pagN* (cell adhesion and invasion [*Lambert and Smith, 2008*; *Folkesson et al., 1999*]) and *pgtE* (virulence [*Hammarlöf et al., 2018*; *Herrero-Fresno et al., 2014*]), were found in 1303 (100%, n=1303/1303) and 1302 (99.9%, n=1302/1303) genomes respectively, and no SNPs in the promoter region of *pgtE* were detected, a mutation previously reported in invasive *S.* Typhimurium ST313 (*Hammarlöf et al., 2018*). In contrast, ST313-gi of unknown function (*Hammarlöf et al., 2018*; *Herrero-Fresno et al., 2014*), the Gifsy-2 like prophage (unknown function [*Hammarlöf et al., 2018*; *Herrero-Fresno et al., 2014*]), and *ggt* gene thought to be involved in gut colonisation (*Mohammed et al., 2017*) were detected in the major ST10 *S.* Dublin lineages, and absent in ST74 *S.* Dublin. The *viaB* operon encoding the Vi capsular antigen was only detected in SLV ST73, a historical isolate from France that displayed the Du3 electrophoretic type that was consistent with a previous study (*Selander et al., 1992*). This suggests that the Vi antigen is not conserved in the *S.* Dublin population or associated with virulence.

The presence of SPI-6 and SPI-19 differed across *S.* Dublin isolates. Both SPIs encode T6SSs that provide a means for injecting effector proteins into adjacent competing bacteria or host cells for immune evasion (*Blondel et al., 2009*; *Sabbagh et al., 2010*). SPI-19 was absent in the ST74 *S.* Dublin lineage but present in all ST10 *S.* Dublin (*Figure 4—figure supplement 1*). The genetic differences

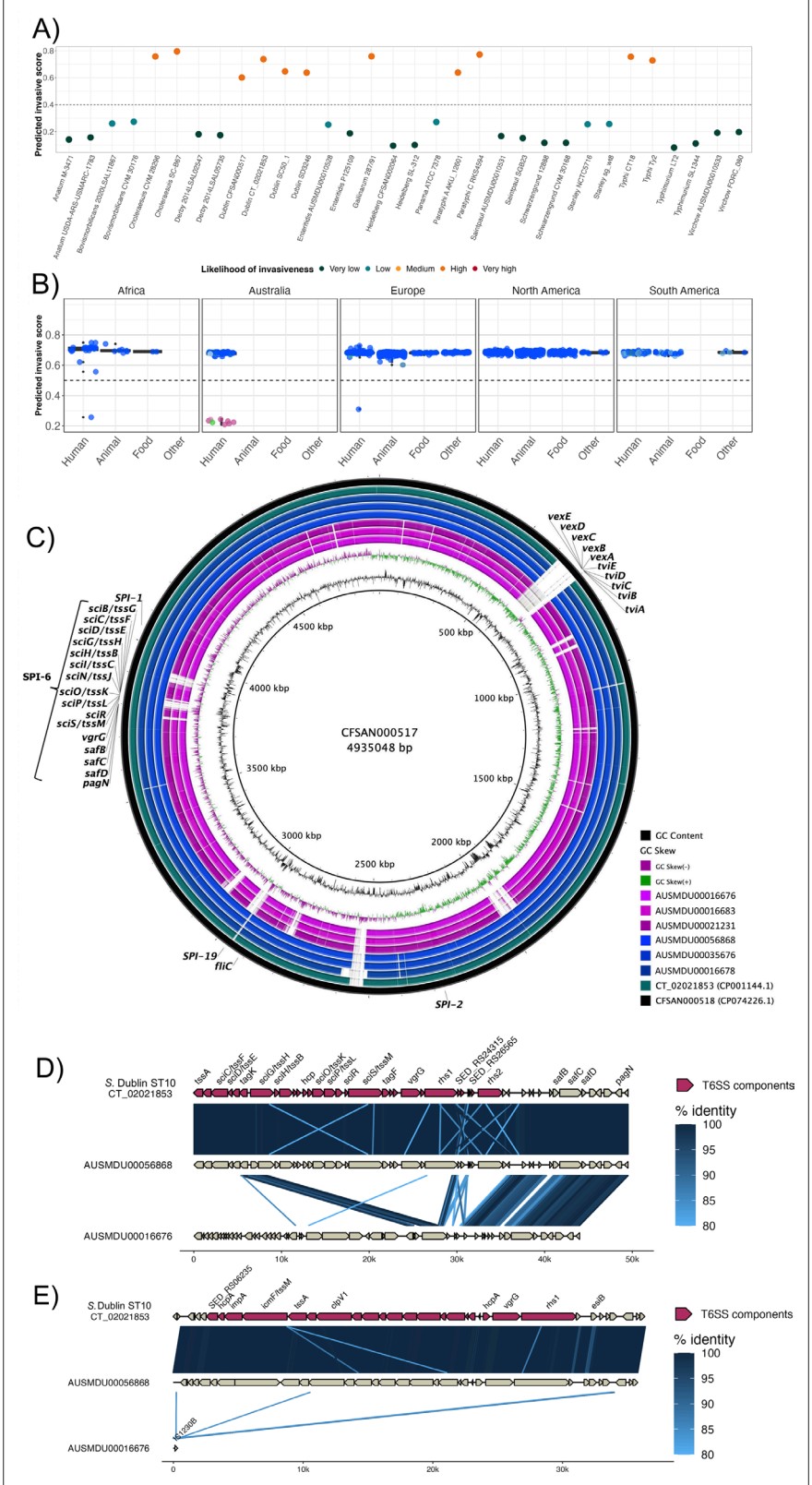

**Figure 4.** Invasive factors associated with *S*. Dublin genomes. Correlation of *Salmonella* pathogenicity islands (SPIs) to *S*. Dublin's invasiveness. (**A**) Validation of the likelihood of invasiveness prediction tool using a variety of publicly available genomes consisting of non-invasive and invasive *Salmonella* serovars (***Supplementary file 1H***). (**B**) The correlation of index scores to the source and its sequence type based on the region. * indicates

*Figure 4 continued on next page*

*Figure 4 continued*

a statistically significant p-value between the index scores of two sources. (**C**) Chromosomal assemblies of sequence type 10 (ST10) (blue) and ST74 (pink) genomes aligned to *S*. Dublin chromosomal reference CT_02021853 (CP001144.1) with a lower limit of 80% identity. (**D**) Aligned sequence of SPI-6 region of an ST10, AUSMDU00056868, and an ST74, AUSMDU00016676, in comparison to SPI-6 region of reference CT_02021853. (**E**) Aligned sequence of SPI-19 region of an ST10, AUSMDU00056868, and an ST74, AUSMDU00016676 in comparison to SPI-19 region of reference CT_02021853.

The online version of this article includes the following figure supplement(s) for figure 4:

**Figure supplement 1.** Population structure of *S*. Dublin genomes with genotypic invasive profiles.

**Figure supplement 2.** Replication dynamics and host cell cytotoxicity induced by *S*. Dublin ST74 and sequence type 10 (ST10) lineage isolates in HT-29 cells.

**Figure supplement 3.** Dot plots comparing bacterial replication and cell death induced by *S*. Dublin ST74 and sequence type 10 (ST10) lineage isolates in THP-1 cells.

**Figure supplement 4.** Pangenome visualisation of gene content of sequence type 10 (ST10) and ST74 lineages.

---

between ST74 and ST10 were reflected in the disparity of their index scores of <0.5 and >0.5, respectively. The SPI-6 region was detected in all isolates; however, <90% sequence coverage was detected in 422/1303 genomes (32.4%). Long-read sequencing on three genomes of the ST74 lineage and three genomes from the ST10 clade 1 lineage corroborated these findings (*Figure 4C–E*), demonstrating truncation of SPI-6 with the T6SS machinery (*Figure 4D*). Long-read sequencing also revealed an absence of SPI-19 in the three ST74 isolates and presence of SPI-19 in the three ST10 isolates (*Figure 4C and E*). The absence of SPI-19 in ST74 suggests that this deletion event occurred during its divergence from the primary *S*. Dublin lineage, ST10.

## Infection of human macrophages reveals distinct intracellular replication patterns between ST10 and ST74 lineages

Given the distinct genomic profiles of ST10 and ST74 lineages, we compared the phenotypic profile of a selection of Australian isolates from each lineage in vitro. The immortalised human macrophage cell line, THP-1, was used to assess invasion, intracellular replication, and host cytotoxicity induced by a subset of clinical isolates from the two lineages. Specifically, we assayed an ST10 population comprising 11 isolates, including an isolate from the highly related ST4293 lineage, and an ST74 population comprising 7 isolates, including an isolate from the highly related ST1545 lineage (*Supplementary file 1J*). Although predicted invasive index scores were higher for the ST10 population (*Figure 4B*), each isolate failed to replicate more than twofold over 24 hr in THP-1 cells (*Figure 5A*), whereas isolates within the ST74 population displayed a sustained increase in fold change of intracellular replication, up to eightfold, over 24 hr (*Figure 5B*). Comparing bacterial growth (CFU/ml) across both sequence types (STs), both lineages were comparable for initial bacterial uptake (*Figure 5C*); however, the ST74 population replicated to significantly higher levels in THP-1 macrophages at 9 and 24 hr post-infection (hpi) compared to the ST10 population (*Figure 5D–F*). Given that NTS also replicate in epithelial cells, we utilised an immortalised human intestinal epithelial cell line, HT-29, to assess *S*. Dublin replication. In contrast to the results obtained in macrophages, we found that 6 out of 11 isolates within the ST10 population replicated more efficiently (up to 10-fold) in HT-29 cells, whereas all 7 isolates within the ST74 population showed minimal replication (up to twofold) over 24 hr (*Figure 4—figure supplement 2A–C*).

## Increased replication of ST74 isolates in human macrophages does not correlate with increased host cell death

It is well documented that prototype *S*. Typhimurium (ST19) potently induces macrophage cytotoxicity both in vitro and in vivo (*Chen et al., 1996*). Here, we utilised a cellular lactate dehydrogenase (LDH) release assay to assess THP-1 cytotoxicity in response to infection with *S*. Dublin ST74 and ST10 populations. Surprisingly, the substantial intracellular replication of the ST74 population in THP-1 cells induced significantly less host cell cytotoxicity at 3 and 9 hpi compared with the ST10 population, suggesting potential immune evasion by ST74 (*Figure 6A and B*). By 24 hpi, however, cytotoxicity induced by both ST74 and ST10 populations was relatively high, with no significant difference

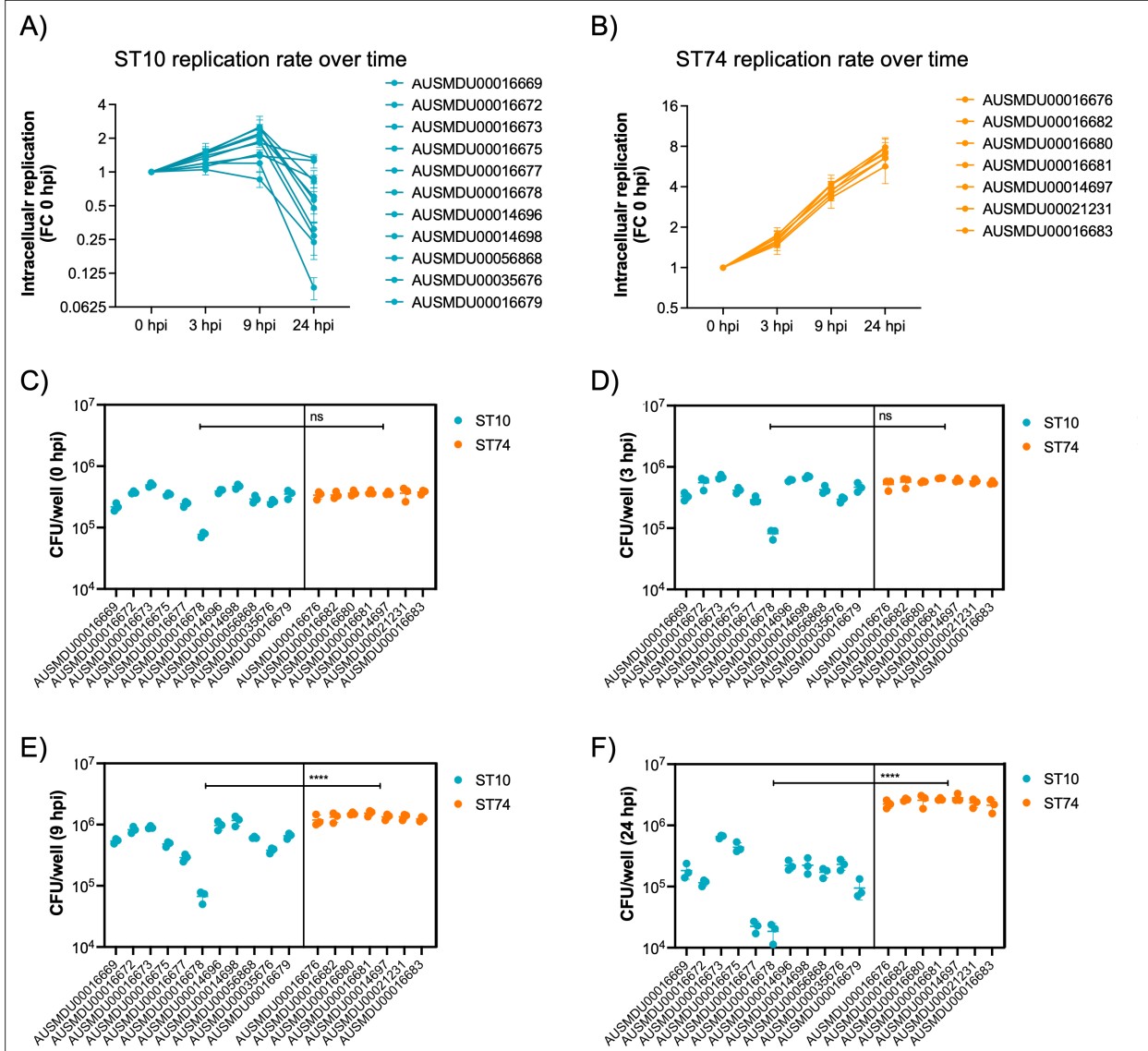

**Figure 5.** Dublin ST74 isolates demonstrate increased intracellular replication in human macrophages over sequence type 10 (ST10) isolates. Differentiated THP-1 (human monocyte) cells were infected at an MOI of 10 with selected ST10 and ST74 *S.* Dublin isolates. (**A, B**) THP-1 cells were lysed, and intracellular bacteria enumerated as fold change in replication and (**C–F**) CFU/well at time 0, 3, 9, and 24 hpi. For (**A**), each measurement at times 3, 9, and 24 hpi represents the fold change in CFU/well compared to 0 hpi, the start of infection, recorded as a biological replicate (performed in technical triplicate), with error bars indicating ±1 standard deviation of n=3 biological replicates. For (**C–D**), each dot represents CFU/well of a biological replicate (performed in technical triplicate), with error bars indicating ±1 standard deviation of n=3 biological replicates. Statistical significance was determined by nested Student's t-test. MOI, multiplicity of infection, hpi, hours post-infection, CFU, colony-forming units, FC, fold change.

observed across the STs (*Figure 6C*). In contrast, there was minimal cytotoxicity observed in infected HT-29 cells across the board (*Figure 4—figure supplement 2D*), suggesting a less potent activation of host cell death in infected epithelial cells compared with macrophages.

To understand how the ST74 population may be suppressing macrophage death at earlier stages of infection, we measured protein secretion of IL-1β via cytokine release and immunoblot assays at 9 hpi, respectively. IL-1β is a potent pro-inflammatory cytokine that is released upon activation of the NLRC4 inflammasome by components of the *Salmonella* type III secretion apparatus (*Miao et al., 2010*), resulting in a lytic cell death process known as pyroptosis (*Bergsbaken et al., 2009*). Here, we observed significantly less IL-1β release by THP-1 cells infected with isolates from the ST74 population at 9 hpi compared with those from the ST10 population (*Figure 6D*), directly reflecting the LDH cytotoxicity data represented in *Figure 6B*. To further support our observations, we selected

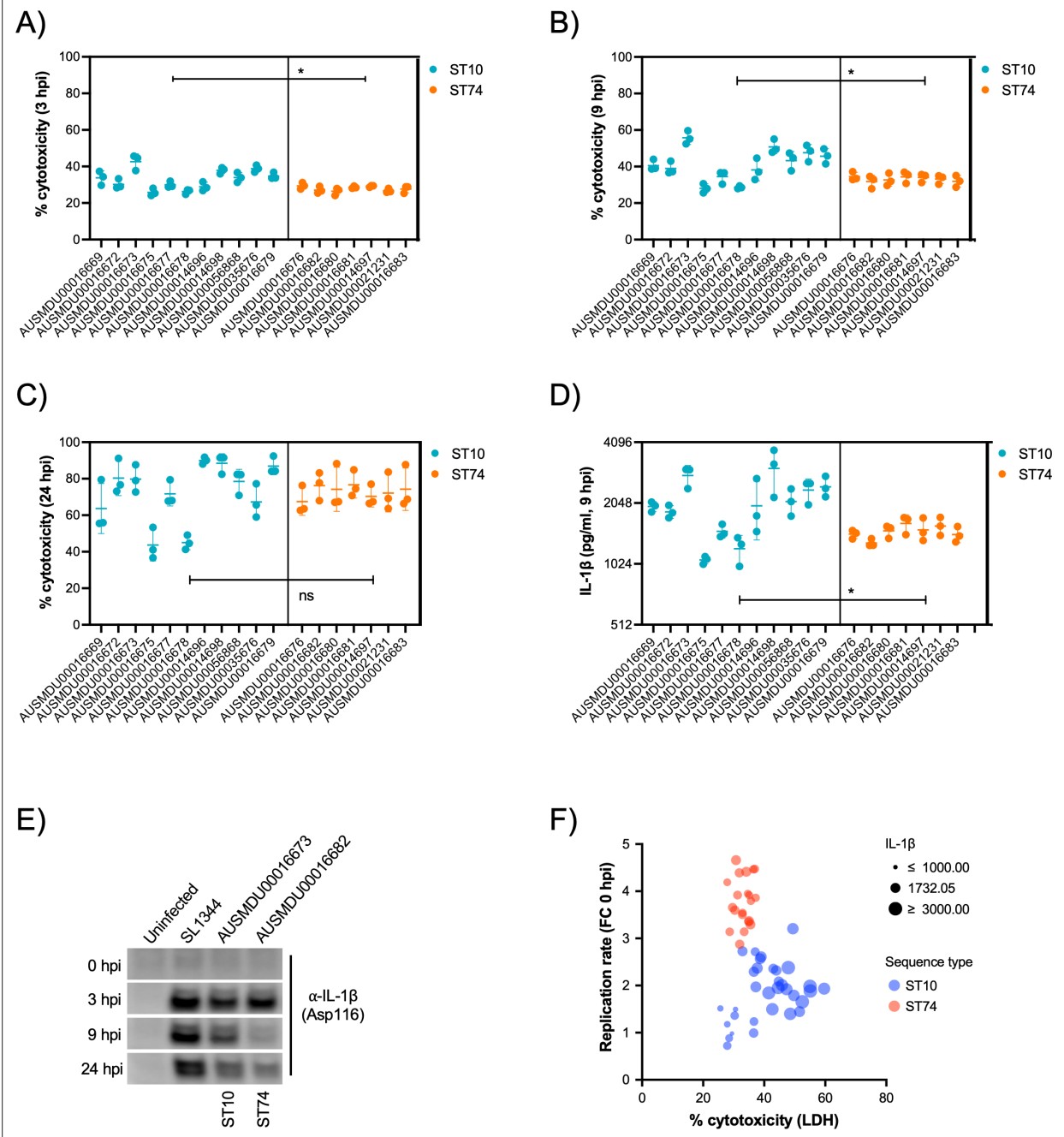

**Figure 6.** *S*. Dublin ST74 isolates induce comparable cytotoxicity to sequence type 10 (ST10) isolates in human macrophages, despite increased intracellular growth. (**A–C**) Cell supernatants were assayed for LDH as a measure of cytopathic effects of infection, and % cytotoxicity was calculated by comparison with 100% lysed uninfected control cells. Samples were collected from THP-1 cells at 3, 9, and 24 hpi. Each point represents the % cytotoxicity of a biological replicate (performed in technical triplicate), with error bars indicating ±1 standard deviation of n=3 biological replicates. Statistical significance was determined by nested Student's t-test. (**D**) IL-1β secretion from THP-1 cells infected for 9 hr with *S*. Dublin isolates. Error bars indicate ±1 standard deviation of n=3 biological replicates and statistical significance determined by Student's t-test. (**E**) Immunoblot of cleaved IL-1β (Asp166) in THP-1 cell lysates infected with *S*. Typhimurium reference strain SL1344, *S*. Dublin ST10 (AUSMDU00016673), or *S*. Dublin ST74 (AUSMDU00016682) over 0, 3, 8, and 24 hr. (**F**) Scatter plot comparing replication rate, % cytotoxicity, and IL-1β secretion from THP-1 cells infected with *S*. Dublin isolates at 9 hpi. Each dot represents each biological replicate of three (averaged from three technical replicates). MOI, multiplicity of infection, hpi, hours post-infection, CFU, colony-forming units, LDH, lactate dehydrogenase.

The online version of this article includes the following source data for figure 6:

**Source data 1.** Uncropped TIFF image for **Figure 6E** immunoblots.

*Figure 6 continued on next page*

*Figure 6 continued*

**Source data 2.** Uncropped TIFF image for *Figure 6E* in PowerPoint format with labelling of cleaved caspase-8 bands and time course of infection indicated.

single representative isolates of ST74 and ST10 to perform immunoblot analyses of cleaved IL-1β (activated form) within THP-1 cells. Here, ST74 induced less IL-1β cleavage at 9 hpi compared to ST10 (*Figure 6E*). We included the ubiquitous laboratory NTS reference strain, *S.* Typhimurium SL1344 in the immunoblot analysis as this is known to potently induce pyroptosis in human macrophages (*Doerflinger et al., 2020*). As expected, SL1344 induced significant IL-1β cleavage during infection of THP-1 macrophages (*Figure 6E*). To assess the correlation of cellular cytotoxicity as measured by LDH, IL-1β release, and intracellular replication, we generated a scatter plot of all isolates within the ST74 and ST10 populations. Here, we can clearly demonstrate that isolates from the ST74 population have increased replicative ability, induce less cytotoxicity, and lower levels of IL-1β release (*Figure 6F*) compared with those within the ST10 population. There were, however, some ST10 isolates that induced similarly low levels of host cell death, presumably due to lower levels of intracellular replication (*Figure 6F*, *Figure 4—figure supplement 3A and B*). These isolates generally clustered away from the ST10 isolates that induced more cell death (*Figure 4—figure supplement 3C*).

A pangenome analysis of the isolates from both ST10 and ST74 lineages revealed that out of a total of 5792 genes, there were 473 and 1143 genes specific to the ST10 and ST74 lineage, respectively (*Figure 4—figure supplement 4*, *Supplementary file 1K*). Further, out of 1143 genes, there were 362 core genes present in all isolates from the ST74 lineage (*Supplementary file 1K*). To highlight these differences, an SNP distance matrix was generated, revealing a minimum of 2402 SNPs between the ST10 and ST74 lineages (*Supplementary file 1L*). While these findings require further investigation, an increased gene content and genome size were consistently observed in all isolates of the ST74 lineage (*Supplementary file 1K and M*). These results indicate a potential link between the unique genome content of the ST74 lineage and alterations in host responses and replicative ability.

## Discussion

Our data provide comprehensive insights into the emergence and spread of *S.* Dublin, with distinct geographical distribution related to AMR and virulence. Analysing the largest global dataset of *S.* Dublin genomes to date, we show that highly resistant *S.* Dublin is almost entirely confined to North America, with emergence in the late 20th century and ongoing local microevolution. Consistent with previous studies, the North American *S.* Dublin isolates in clade 5 were characterised by the presence of an IncC plasmid, often with co-location of AMR and HMR determinants (*Hsu et al., 2019*; *Mangat et al., 2017*), while the clades circulating in Europe were associated with IncN plasmids (*Kudirkiene et al., 2020*). Previous studies have explored these IncC plasmids in detail. The use of biocides and heavy metals in the agri-food industry may facilitate co-selection, contributing to the persistence of AMR (*Cheng et al., 2019b*). This highlights the potential for selection pressure in one ecological niche (e.g. the use of biocides in farming or heavy metals in animal feeds) leading to AMR in another (e.g. AMR in humans).

The phylogeographic separation, AMR, and plasmid profiles of *S.* Dublin lineages are consistent with other studies (*Mangat et al., 2019*; *Fenske et al., 2019*; *Srednik et al., 2021*; *Fritz et al., 2022*; *McMillan et al., 2020*; *Zhang et al., 2020*; *Izzo et al., 2011*), and are likely to be influenced by several factors, including local AMR use and patterns of livestock movement. Given the livestock reservoir of *S.* Dublin is mostly host-restricted to cattle, it is conceivable that both historic and contemporary global cattle movement may have contributed to region-specific differences in *S.* Dublin. For example, Australian *S.* Dublin genomes in clade 1 shared a common ancestor with North American isolates in clade 5. During the early 1930s, Brahman cattle were imported from the USA into Australia (*O'Connell, 1930*), supporting our phylodynamic finding that the common ancestral link between *S.* Dublin isolates from North America and Australia emerged in the early to mid-20th century. Following region-specific introductions, subsequent local antimicrobial and husbandry practices may have contributed to the local evolution of *S.* Dublin (*Fritz et al., 2022*; *Schroll et al., 2019*).

Of note, among *S.* Dublin isolates from Australia, genomic analysis (cf. serotyping) revealed a smaller population of ST74 isolates. *S.* Dublin ST74 was previously identified as *S.* Enteritidis ST74

and sits as an intermediate between *S.* Dublin and *S.* Enteritidis (*Achtman et al., 2012*), and has been understudied to date in studies of *S.* Dublin. The difference between the antigenic formula of *S.* Dublin and *S.* Enteritidis is in the H1 flagella region, which is typed as '*g,p*' for *S.* Dublin and '*g,m*' for *S.* Enteritidis. ST74 *S.* Dublin genomes were characterised by: (i) a lack of any AMR or HMR determinants; (ii) lack of complete SPI-6 regions, specifically the T6SS, and (iii) a lack of putative virulence determinants, including ST313-gi, Gifsy-2-like prophage, and SPI-19, which encodes a T6SS mechanism for intestinal colonisation and survival that interacts with the host by mediating macrophage cytotoxicity (*Schroll et al., 2019*).

The lack of the Vi antigen in 1302/1303 (99.9%) *S.* Dublin genomes has several implications. First, the antigenic formula of *S.* Dublin, 1,9,12[Vi] may need to be reviewed if most *S.* Dublin genomes lack the machinery to express the Vi capsule. These limitations with serotyping have been highlighted in a previous study, which suggested a transition to a novel genotyping scheme that would prevent these phenotypic discrepancies (*Chattaway et al., 2021*). Second, the Vi antigen is a vaccine target for *S.* Typhi (*Hu et al., 2017*), and the Vi antigen has been explored as a target for *S.* Dublin (*Janis et al., 2011*). While previous studies have noted that some *S.* Dublin genomes may be capable of producing the Vi antigen (*Janis et al., 2011*), the genomic data from this study shows that these genomes are in the minority of the *S.* Dublin population and may not be stable targets for *S.* Dublin-specific vaccines.

Our phenotypic data demonstrated a striking difference in replication dynamics between ST10 and ST74 populations in human macrophages. ST74 isolates replicated significantly over 24 hr, whereas ST10 isolates were rapidly cleared after 9 hr of infection. ST74 induced significantly less host cell death during the early to mid-stage of macrophage infection, supported by limited processing and release of IL-1β at 9 hpi. While NTS are generally potent inflammasome activators (*Clare, 2021*), most data that support this observation have been generated using laboratory-adapted *S.* Typhimurium strains, often ST19. Our findings suggest that ST74 isolates may employ immune evasion mechanisms to avoid host recognition and activation of cell death signalling in the early stages of infection. Similar trends have been observed with other invasive *S. enterica* serovars, including *S.* Typhimurium ST313, which induces less inflammasome activation than ST19 during murine macrophage infection (*Carden et al., 2015*). This suppression of inflammatory cell death may facilitate increased replication and dissemination of ST74 isolates at later stages of infection. Consistent with this, we observed comparable cytotoxicity between ST10 and ST74 populations at 24 hpi, suggesting ST74 induces cell death via alternative mechanisms once intracellular bacterial numbers are unsustainable. Further research is needed to identify genomic factors underpinning these observations.

In this study, ST74 isolates associated with human disease (n=7) were identified from faecal samples (cases of gastroenteritis), whereas 18/40 ST10 isolates were from invasive cases of human disease (from blood or abscess). In line with this, our machine learning approach predicted lower invasiveness for ST74 compared to ST10. However, this does not align with what we observed in vitro, i.e., higher replication of ST74 in macrophages compared with ST10. We speculate that the increased genomic complexity of ST74 may support higher replication in macrophages, and that increased intracellular replication may enhance systemic dissemination, though this would require robust in vivo validation. Invasiveness of *S. enterica* is often linked to genome degradation (*Kingsley et al., 2009*; *Feasey et al., 2016*; *Pulford et al., 2021*; *Zhou et al., 2023*). However, this is mostly based on studies of human-adapted iNTS (ST313) and *S.* Typhi, leaving open the possibility that the additional genomic richness of ST74 supports survival in diverse host species. An uncharacterised virulence factor or SNP within virulence genes of ST74 may also explain this replication advantage. Interestingly, the absence of SPI-19 in ST74, which encodes a T6SS, may reflect adaptation to enhanced replication in macrophages. SPI-19 has been linked to intestinal colonisation in poultry (*Pezoa et al., 2014*; *Schroll et al., 2019*) and mucosal virulence in mice (*Schroll et al., 2019*). It is possible that the efficient replication of ST74 in macrophages may compensate for the absence of SPI-19, relying instead on phagocyte uptake via M cells or dendritic cells. Collectively, these findings highlight phenotypic differences between *S.* Dublin populations ST10 and ST74, whereby enhanced intra-macrophage survival of ST74 could promote invasive disease, whereas the prevalence of ST10 may relate to better intestinal adaptation and enhanced faecal shedding. These findings highlight important knowledge gaps in zoonotic NTS host-pathogen interactions and drivers of emerging invasive NTS lineages with broad host ranges.

This study has limitations, including a focus on ST10 isolates from clade 1, which do not represent global phylogenetic diversity. Nonetheless, our pangenome analysis identified >900 uncharacterised

genes unique to ST74, offering potential targets for future research. Another limitation is the geographical bias in available genomes, with underrepresentation from Asia and South America. This reflects broader disparities in genomic epidemiological studies but may improve as public health genomics capacity expands globally.

Overall, this study represents the most comprehensive genomic study of *S.* Dublin to date, systematically characterising antimicrobial, heavy metal, and biocide resistance determinants, in addition to exploring the virulome in the global population of *S.* Dublin. Importantly, our phenotypic data reveals the caution required when making generalised statements about infection outcomes and virulence mechanisms of *Salmonella* serovars. Our data clearly shows that within the Dublin serovar, there are distinct phenotypic differences observed across STs and host cell types, all of which could impact significantly on future development of effective therapeutics or vaccine strategies for NTS infections. While the data did not indicate an increasing trend of iNTS associated with *S.* Dublin, the potential public health risk of this invasive pathogen suggests it may still warrant considering it a notifiable disease, similar to typhoid and paratyphoid fever.

## Methods

### Setting and dataset

For Australian isolates, whole-genome sequencing (WGS) was performed on all viable *S.* Dublin isolates from human clinical samples received at the Microbiological Diagnostic Unit Public Health Laboratory (MDU PHL), the primary bacterial reference laboratory for the State of Victoria in Australia. In total, 53 *S.* Dublin genomes between 2000 and 2019 were included. For publicly available data, we included *S.* Dublin sequences that had associated metadata (geographical origin and date of sample collection), providing a total of 1250 publicly available sequences. Data were collected in accordance with the Victorian Public Health and Wellbeing Act 2008. Ethical approval was received from the University of Melbourne Human Research Ethics Committee (study number 1954615.3).

### Genome quality control, assembly, and serovar prediction

Genomes were assembled and quality controlled using the Nullarbor pipeline v2.0.20191013 (https://github.com/tseemann/nullarbor; *Seemann, 2020a*). Briefly, Skesa v2.4.0 (*Souvorov et al., 2018*) was used to assemble the genomes before determining the ST using the multilocus sequence typing (MLST) scheme 'senterica' within mlst v2.19.0 (https://github.com/tseemann/mlst; *Seemann, 2020b*). Kraken v1.1.1 (*Wood and Salzberg, 2014*) was then used to confirm the taxonomic classification. Snippy v4.6.0 (https://github.com/tseemann/snippy; *Seemann, 2020c*) with parameters set to a 0.90 variant rate proportion against a minimum coverage of 10× mapped the reads against a well-studied reference genome, *S.* Dublin CT_02021853 (NC_011205.1). To be included in subsequent analysis, genomes had to pass the following quality control parameters: mapped to ≥80% of the reference genome; had a ≥80% taxonomic sequence similarity to *S. enterica*; N50 >25,000, and <300 contigs. To verify that each genome was *S.* Dublin, all sequences were typed using *Salmonella* in silico Typing Resource (SISTR) v1.1.1 (*Yoshida et al., 2016*). The electrophoretic type of each isolate was determined using Antimicrobial Resistance Identification by Assembly (ARIBA) v2.14.5 (*Hunt et al., 2017*) against the reference Du 2 (Genbank accession: M84972.1). The draft genome assemblies were annotated using Prokka v1.14.6 (*Seemann, 2014*).

### Core genome phylogenomic analysis

Variant calls from snippy were used to construct a core genome SNP alignment using snippy-core v4.6.0 (https://github.com/tseemann/snippy; *Seemann, 2020c*). Phaster v2020-12-22 (*Arndt et al., 2016*) was used to create a BED file containing prophage regions. The option '—mask' in snippy-core was used to exclude the identified prophage regions from the BED file. Recombination was assessed and filtered using Gubbins v2.4.1 (*Croucher et al., 2015*). Constant sites from the reference genome were obtained using the script iqtree-calc_const_sites.sh (https://github.com/MDU-PHL/mdu-tools/blob/master/bin/iqtree-calc_const_sites.sh; *Seemann, 2021a*). An ML phylogeny was inferred using IQ-TREE v2.1.0 (*Nguyen et al., 2015*) with parameters set to a GTR+G4 nucleotide substitution model with constant sites and ultrafast bootstrapping (*Minh et al., 2013*) to 1000 replicates. The resulting ML tree was then visualised using *ggtree* v2.3.3 R package (*Yu et al., 2017*). Population clades were

defined using rhierBAPS v1.1.3 (*Tonkin-Hill et al., 2018*) to one level on an alignment of 2957 core SNPs of 1295 isolates in the ST10 *S.* Dublin lineage.

## In silico prediction of resistance determinants and plasmid replicons

To screen for quinolone point mutations and acquired AMR determinants, ARIBA and AbriTAMR v0.2.2 (https://github.com/MDU-PHL/abritamr; *Horan et al., 2021*). were used, respectively. The AMR database used for this study was the NCBI AMRFinderPlus database v2020-01-22.1. Each isolate's susceptibility profile was classified as either susceptible, resistant to one to two antimicrobials, or MDR. Heavy metal and biocide determinants were identified by screening the AMRFinder database v2020-07-16.2 with AMRFinder v3.8.4 (*Feldgarden et al., 2019*). Replicon types were determined using ABRicate v1.0.1 (https://github.com/tseemann/abricate; *Seemann, 2020d*) at a threshold of 80% nucleotide identity and 95% coverage, and screening against the PlasmidFinder database (*Carattoli et al., 2014*). A strict threshold was applied to ensure the detection of only unambiguous replicon types. To establish the most common resistant profiles, co-occurrence matrices of resistant determinants were constructed in R using *igraph* v1.2.4.1 (*Csardi and Nepusz, 2006*).

## Virulome analysis

Virulence genes were identified using ABRicate at 90% nucleotide identity/coverage against the virulence factor database (VFDB) v2018-03-18 (*Chen et al., 2016*). Databases from previous studies (*Hsu et al., 2019*; *Hammarlöf et al., 2018*; *Mohammed et al., 2017*; *Mansour et al., 2020*; *Suez et al., 2013*; *Sévellec et al., 2018*; *Hayward et al., 2013*; *Kérouanton et al., 2015*) were also interrogated and collated. This included a database containing additional virulent determinants and SPIs previously implicated with *S.* Dublin invasome (*Mohammed et al., 2017*; *Herrero-Fresno et al., 2014*). Using ABRicate, the presence of SPIs -1, -2, -6, and -19 was identified at 80% ID and 10% COV (*Supplementary file 1N*). ABRicate was also used to screen the manually created SPI determinant database for genes relevant to SPI-1, -2, -6, and -19 with a threshold of 90% identity/coverage (*Supplementary file 1N*). To quantify invasiveness, index scores were calculated in silico for each isolate using Wheeler et al.'s tool developed in 2018 for inferring the likelihood of invasiveness (https://github.com/Gardner-BinfLab/invasive_salmonella; *Wheeler, 2023*) with a higher score for an isolate deemed more invasive (*Wheeler et al., 2018*). Prior to running the tool, a set of reference genomes consisting of invasive and non-invasive *Salmonella* serovars were used as a validation set (*Supplementary file 1H*). The validation results were then visualised using *ggtree*.

## Bayesian phylodynamic analysis

A subset of 660 *S.* Dublin genomes from the main cluster were used for BEAST v1.10.4 (*Suchard et al., 2018*) analysis. This subset comprised all Australian genomes and a proportion of international *S.* Dublin genomes. The sampling strategy for selecting representative international genomes involved capturing all geographical regions and sources harbouring distinct susceptibility profiles. When two genomes demonstrated the same AMR profile and were from the same region and source, the oldest genome was preferred. Recombination sites were filtered using Gubbins. A core genome SNP alignment of these genomes was created using snippy-core v4.6.0, and then an ML phylogeny was inferred as above. Temporal signal was assessed using TempEst v1.5.3 (*Rambaut et al., 2016*) by assessing the best-fitting root using the heuristic residual mean squared method, with an $R^2$ of 0.31. The core alignment was then used to create an XML file in BEAUti v1.10.4 using the GTR+G4 substitution model. To identify the most appropriate model for our dataset, a combination of using either a 'relaxed lognormal' or a 'strict' molecular clock with either a 'coalescent constant' or a 'coalescent exponential' population prior was tested. For all models, the Markov chain Monte Carlo chain length was set to 300,000,000, and sampling was set to every 20,000 trees with the marginal likelihood estimation set to 'generalised stepping stone (GSS)' to assess the statistical support for models and evaluate the best-fit model. The analysis was repeated in triplicates with and without tip dates to ensure there was no temporal bias. Bayesian evaluation of temporal signal was used to determine whether a temporal signal was present (*Duchene et al., 2020*). To optimise computational resource and time, BEAST v1.10.4 was used with BEAGLE v3.0.2 (*Ayres et al., 2012*). An MCC tree from the highest supported model using GSS was extracted using TreeAnnotator v1.10.4 and median heights. The resulting MCC tree was then visualised using *ggtree*.

## Long-read sequencing

The Oxford Nanopore Technologies MinION platform was used to obtain complete genomes of six Australian *S.* Dublin genomes (*Supplementary file 1M*). These genomes, comprised of ST10 lineage clade 1 (AUSMDU00056868 and AUSMDU00035676) and ST74 (AUSMDU00016676 and AUSMDU00021231), were selected based on their AMR profile, virulome, and index scores. To prepare new DNA libraries for long-read sequencing, genomes were freshly re-cultured. DNA was extracted using the Virus/Pathogen DSP midi kit on the QiaSymphony SP instrument. Sequencing libraries were prepared using a Ligation sequencing kit (SQK-LSK109) with native barcode expansion and sequenced for 48 hr on a GridION Mk2 instrument using an R9.4 flowcell. WGS was performed using the Illumina NextSeq 500/550 platform (Illumina, San Diego, CA, USA). Demultiplexed long reads and corresponding WGS reads were subjected to QC to exclude unreliable reads using fastp v0.20.1 (*Chen et al., 2018*) and Filtlong v0.2.1 (https://github.com/rrwick/Filtlong; *Wick, 2021*) respectively. Long reads were assembled using Trycycler v0.4.1 (*Wick et al., 2021*). The final sequence was polished using Medaka v1.4.3 (https://github.com/nanoporetech/medaka; *Oxford Nanopore Technologies, 2021*). To ensure small plasmids were not excluded during the initial QC step, hybrid assemblies using both long and WGS reads were generated with Unicycler v0.4.4 (*Wick et al., 2017*). Trycycler assemblies were checked against Unicycler assemblies with Bandage v0.8.1 (*Wick et al., 2015*). The final genome assemblies were confirmed as *S.* Dublin using SISTR and annotated using both Prokka v1.14.6 (*Seemann, 2014*) for consistency with the draft genome assemblies and Bakta v1.10.1 (*Schwengers et al., 2021*), which provides for more detailed annotations (*Supplementary file 1M*). Both Prokka and Bakta annotations were in agreement for AMR, HMR, and virulence genes, with Bakta annotating between three and seven additional coding sequences, which were largely 'hypothetical protein'.

Chromosomal assemblies were aligned to complete genomes of known *S.* Dublin references CT_02021853 (CP001144.1, without Vi capsule) and CFSAN000518 (CP074226.1, with Vi capsule) and visualised using Blast Ring Image Generator (BRIG) v0.95 (*Alikhan et al., 2011*). Plasmid assemblies were screened against the PlasmidFinder database (*Carattoli et al., 2014*) and mob-typer v3.1.0 (*Robertson and Nash, 2018*) to identify replicon and relaxase types, respectively. Plasmid profiles were subjected to BLASTn against previously identified IncX1/IncFII(S) virulence plasmids reported in *S.* Dublin and IncN plasmids associated with serovars of *Salmonella* (*Supplementary file 1E*). Genome comparisons of plasmids demonstrating the top four highest BLASTn max scores were visualised using BRIG. Bakta, PlasmidFinder, and ISFinder (*Siguier et al., 2006*) were used to annotate the regions of interest, while Easyfig v2.2.2 (*Sullivan et al., 2011*) was used to compare regions of the genome. Regions of the genome were visualised using the R package *gggenomes* v0.9.12.9000 (https://github.com/thackl/gggenomes; *Hackl et al., 2024*).

## Human cell line infections

The human macrophage cell line (THP-1) (ATCC TIB-202) and human colonic epithelial cell line (HT-29) (ATCC HTB-38) were obtained from the American Type Culture Collection (ATCC). Both cell lines were authenticated by the ATCC upon acquisition. The identity of the cell lines was regularly verified through Short Tandem Repeat profiling. Additionally, these cell lines were tested for mycoplasma contamination every 3 months, and all tests returned negative results. THP-1 and HT-29 cell lines were maintained in Roswell Park Memorial Institute (RPMI) 1640 media + 200 mM GlutaMAX (Life Technologies) supplemented with 10% vol/vol fetal bovine serum (FBS; Bovogen) and grown in a humidified 5% $CO_2$ 37°C incubator. THP-1 monocytes were differentiated into macrophages with 50 ng/mL phorbol 12-myristiate-12 acetate (Sigma-Aldrich) for 72 hr prior to infection.

Single colonies of each *S.* Dublin isolate were inoculated into 10 mL Luria Bertani (LB) broth and grown overnight at 37°C at 200 rpm, then subcultured for 3 hr in the same conditions in 5 mL LB. Colony-forming units/mL were measured by absorbance at 600 nM ($OD_{600}$). THP-1 and HT-29 cells were infected at a multiplicity of infection (MOI) of 10 in RPMI in triplicate wells of a 96-well plate and centrifuged at 525 × *g* for 5 min to synchronise bacterial uptake. Infected cells were incubated at 37°C for 30 min, then infective media removed and replaced with 100 µg/mL gentamicin in 10% vol/vol FBS/RPMI to inhibit extracellular bacterial growth. 0 hpi samples were collected at this time to measure initial bacterial uptake. Cells were then incubated in high-dose gentamicin for 1 hr, then replaced with 10 µg/mL gentamicin in 10% vol/vol FBS/RPMI for the remainder of the infection. For

THP-1 cells, samples were collected at time 0, 3, 9, and 24 hpi. For HT-29 cells, samples were collected at 0 and 24 hpi.

Cytotoxicity of infection was measured by LDH release by infected cells before lysis for intracellular bacterial enumeration. Briefly, 35 µL of infected (and uninfected) cell culture supernatants were collected from each well and LDH quantified as per the manufacturer's instructions (Promega), with percent cytotoxicity calculated as a proportion of lysed control cells and blank media. Following this, infected cells were washed three times with warm phosphate-buffered saline (PBS) to remove any lingering extracellular bacteria, then lysed in 0.1% Triton X-100/PBS and plated out onto LB agar after serial dilution. Bacterial enumeration and cell viability assays were performed over three biological replicates, each performed independently, with a new passage of host cells, on a separate day. Within each biological replicate, technical triplicates were performed for each isolate. All statistical analyses were performed using Prism software (GraphPad Software v9.0) and determined by nested Student's t-test, which allowed for inclusion of individual isolates in subgroups of each ST (ST10 vs ST74).

## Measurement of cellular responses

THP-1 cells were infected at an MOI of 10 with a representative isolate from each of the ST10 and ST74 lineages as described above, then lysed in radio-immunoprecipitation buffer (20 mM Tris pH 8.0, 0.5 mM EDTA, 150 mM NaCl, 1% vol/vol Triton X-100, 0.5% wt/vol sodium deoxycholate, 0.1% wt/vol SDS) supplemented with Complete Protease and Phosphatase Inhibitor Cocktail (Roche) at 0, 3, 8, and 24 hpi. Total protein concentration was quantified by bicinchoninic acid (BCA) assay and 15 µg separated on 4–12% Bolt Bis-Tris polyacrylamide gels (Thermo Scientific), transferred to nitrocellulose (iBlot, Thermo Scientific), and blotted for cleaved IL-1β (clone D3A3Z, Cell Signaling). Cell culture supernatants were also collected from 96-well plate infections at 9 hpi, and secreted IL-1β measured by DuoSet Human IL-1β/IL-1F2 ELISA as per the manufacturer's instructions (R&D Systems). A core genome SNP ML phylogeny was constructed on the Australian ST10 lineage to demonstrate the genetic relationship of representative ST10 isolates that underwent in vitro phenotypic assays.

## Pangenome analysis

The pangenomes of the isolates used in the macrophages and epithelial infections were analysed using Panaroo v1.3.4 (*Tonkin-Hill et al., 2020*) set to default threshold settings, under strict mode and with the removal of invalid genes on the Prokka annotated draft genome assemblies (*Supplementary file 1K*). An SNP distance matrix was generated using snp-dists v0.8.2 (https://github.com/tseemann/snp-dists; *Seemann, 2021b*) based on the core genome alignment of these isolates, which had been previously used to construct an ML tree.

## Code availability

The machine learning tool developed by Wheeler et al. for predicting the likelihood of invasiveness is available in a public GitHub depository https://github.com/Gardner-BinfLab/invasive_salmonella (*Wheeler, 2023*).

## Acknowledgements

We thank all staff members of the Enterics and MDDI section at the MDU PHL for their technical assistance. We would also like to thank Jake Lacey for his assistance uploading the reads to NCBI. The BEAST analysis for this paper was undertaken using the LIEF HPC-GPGPU Facility hosted at the University of Melbourne. This Facility was established with the assistance of LIEF Grant LE170100200. DAW was supported by a National Health and Medical Research Council (NHMRC) Investigator Grant (GNT1174555) and a Medical Research Future Fund (MRFF) Grant (FSPGN000045). BPH is supported by NHMRC Investigator Grant (GNT1196103). DJI is supported by an NHMRC Investigator Grant (GNT1195210). JSP was supported by a Sylvia and Charles Viertel Senior Medical Research Fellowship (SMRF22008). MDU PHL is funded by the Victorian Government, Australia. No conflict of interest declared.

## Additional information

### Funding

| Funder | Grant reference number | Author |
|---|---|---|
| National Health and Medical Research Council | GNT1174555 | Deborah A Williamson |
| Sylvia and Charles Viertel Charitable Foundation | SMRF22008 | Jaclyn S Pearson |
| National Health and Medical Research Council | GNT1196103 | Benjamin P Howden |
| National Health and Medical Research Council | GNT1195210 | Danielle J Ingle |
| Australian Government | Medical Research Future Fund FSPGN000045 | Deborah A Williamson |

The funders had no role in study design, data collection and interpretation, or the decision to submit the work for publication.

### Author contributions

Cheryll M Sia, Rebecca L Ambrose, Conceptualization, Data curation, Formal analysis, Validation, Investigation, Visualization, Methodology, Writing – original draft, Project administration, Writing – review and editing; Mary Valcanis, Patiyan Andersson, Susan A Ballard, Resources, Data curation; Benjamin P Howden, Resources, Data curation, Supervision; Deborah A Williamson, Conceptualization, Resources, Formal analysis, Supervision, Funding acquisition, Visualization, Methodology, Writing – original draft, Project administration, Writing – review and editing; Jaclyn S Pearson, Danielle J Ingle, Conceptualization, Resources, Data curation, Formal analysis, Supervision, Funding acquisition, Validation, Investigation, Visualization, Methodology, Writing – original draft, Project administration, Writing – review and editing

### Author ORCIDs

Cheryll M Sia  http://orcid.org/0000-0002-7058-6022
Benjamin P Howden  https://orcid.org/0000-0003-0237-1473
Jaclyn S Pearson  https://orcid.org/0000-0002-7358-4479
Danielle J Ingle  https://orcid.org/0000-0003-0707-6537

Reviewer #1 (Public review): https://doi.org/10.7554/eLife.102253.4.sa1
Reviewer #2 (Public review): https://doi.org/10.7554/eLife.102253.4.sa2
Author response https://doi.org/10.7554/eLife.102253.4.sa3

## Additional files

### Supplementary files

Supplementary file 1. (**A**) Metadata with genotypic resistance profiles of 1303 *S*. Dublin genomes. (**B**) Distribution of genomes according to source and region. (**C**) Distribution of genomes according to sequence type and region. (**D**) *fliC* and Vi profiles of 1303 *S*. Dublin isolates in study. (**E**) Sequence similarity to reference plasmids. (**F**) Genome annotations of plasmid AUSMDU00035676. (**G**) Genome annotations of plasmid AUSMDU00056868. (**H**) Validation of invasive index prediction tool. (**I**) Metadata with virulome profiles of 1303 *S*. Dublin genomes. (**J**) Representative ST10 and ST74 populations used in phenotypic assays. (**K**) Results from pangenome analysis of representative isolates from ST10 and ST74 lineages. (**L**) Single nucleotide polymorphisms (SNPs) distance matrix of representative isolates from ST10 and ST74 lineages. (**M**) Genomes sent for long-read sequencing assemblies. (**N**) Database of virulent determinants that have been associated with *S*. Dublin.

MDAR checklist

## Data availability

Paired end reads of public data were obtained from either the European Nucleotide Archive (ENA) or National Center for Biotechnology Information (NCBI). Details of accessions and original studies are included in Supplementary file 1A. The Australian unfiltered FASTQ short read data and ONT data generated for this study have been submitted to the NCBI BioProject database under accession number PRJNA319593 with details of the individual accessions in Supplementary file 1A and M. The interactive tree is available in Microreact under https://microreact.org/project/sdublinpaper. The supplementary tables, located within Supplementary file 1, include all data generated in this study. Supplementary file 1A provides full details of the metadata (year, country, region) associated with each genome. It also includes the ST and BAPS clusters, source and sample types, inferred AMR susceptibility profiles, individual AMR mechanisms and presence and absence of plasmid replicons for all 1,303 genomes. These are the source data for Figures 1–3 and Figure 1—figure supplement 1, Figure 3—figure supplement 1, Figure 3—figure supplement 2, and Figure 3—figure supplement 3. Supplementary file 1B provides the counts according to source of isolates and geographical region. Supplementary file 1C provides the counts according to ST and geographical region. Supplementary file 1D has the underlying data for the fliC and Vi antigen profiles of 1,303 genomes. Supplementary file 1E details the previously publicly available virulence and IncN plasmid with the BLAST results linked to Figure 4, Figure 2—figure supplement 1 and Figure 2—figure supplement 2. Supplementary file 1F and G provide the detailed annotations of plasmids underpinning Figure 4, Figure 2—figure supplement 1 and Figure 2—figure supplement 2. Supplementary file 1H provides the source data for Figure 4, showing the invasive index scores of different serovars of *Salmonella*. Supplementary file 1I presents the presence or absence of virulence genes that serve as source data for Figure 4-figure supplement 1. Supplementary file 1J provides the sample type for the isolates used in phenotypic assays and ST. Supplementary file 1K provides the pangenome data used as source data for Figure 4-figure supplement 4. Supplementary file 1L provides the pairwise SNP-distances for the isolates used in the phenotypic analyses. Supplementary file 1M provides the BioSample and AUSMDU IDs for isolates that underwent ONT. Supplementary file 1N provides details of the curated virulence determinants, included in Supplementary file 1I, sourced from two previous studies. Figure 6—source data 1 is the source data for immunoblots for Figure 6E.

The following datasets were generated:

| Author(s) | Year | Dataset title | Dataset URL | Database and Identifier |
|---|---|---|---|---|
| Sia CM, Ambrose RL, Valcanis M, Andersson P, Ballard SA, Howden BP, Williamson DA, Pearson JS, Ingle DJ | 2025 | *S.* Dublin paper | https://microreact.org/project/sdublinpaper | Microreact, sdublinpaper |
| Sia CM, Ambrose RL, Valcanis M, Andersson P, Ballard SA, Howden BP, Williamson DA, Pearson JS, Ingle DJ | 2025 | Whole genome sequencing of *Salmonella enterica* isolates as part of MDU PHL routine surveillance activities | https://www.ncbi.nlm.nih.gov/bioproject/?term=PRJNA319593 | NCBI BioProject, PRJNA319593 |

The following previously published datasets were used:

| Author(s) | Year | Dataset title | Dataset URL | Database and Identifier |
|---|---|---|---|---|
| Kudirkiene E, Sørensen G, Torpdahl M, de Knegt LV, Nielsen LR, Rattenborg E | 2020 | Sequencing of S. Dublin from cattle in Denmark | https://www.ncbi.nlm.nih.gov/bioproject/PRJEB26501 | NCBI BioProject, PRJEB26501 |

*Continued on next page*

*Continued*

| Author(s) | Year | Dataset title | Dataset URL | Database and Identifier |
|---|---|---|---|---|
| Fenske GJ, Thachil A, McDonough PL, Glaser A, Scaria J | 2019 | Bloodstream surveillance across Africas | https://www.ncbi.nlm.nih.gov/bioproject/?term=PRJEB8666 | NCBI BioProject, PRJEB8666 |
| Mangat CS, Bekal S, Avery BP, Côté G, Daignault D, Doualla-Bell F | 2019 | *Salmonella enterica* subsp. enterica serovar Dublin Raw sequence reads | https://www.ncbi.nlm.nih.gov/bioproject/?term=PRJNA402038 | NCBI BioProject, PRJNA402038 |
| Statens Serum Institut | 2019 | *Salmonella* Dublin from Denmark | https://www.ncbi.nlm.nih.gov/bioproject/?term=PRJEB33058 | NCBI BioProject, PRJEB33058 |

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

# Appendix 1

**Appendix 1—key resources table**

| Reagent type (species) or resource | Designation | Source or reference | Identifiers | Additional information |
|---|---|---|---|---|
| Strain, strain background (*Salmonella enterica* serovar Dublin) | ST10 strain used in phenotypic assays - this study. | Microbiological Diagnostic Unit, University of Melbourne | AUSMDU00014696 | Sample type: blood |
| Strain, strain background (*Salmonella enterica* serovar Dublin) | ST10 strain used in phenotypic assays - this study. | Microbiological Diagnostic Unit, University of Melbourne | AUSMDU00014698 | Sample type: faeces |
| Strain, strain background (*Salmonella enterica* serovar Dublin) | ST10 strain used in phenotypic assays - this study. | Microbiological Diagnostic Unit, University of Melbourne | AUSMDU00016669 | Sample type: blood |
| Strain, strain background (*Salmonella enterica* serovar Dublin) | ST10 strain used in phenotypic assays - this study. | Microbiological Diagnostic Unit, University of Melbourne | AUSMDU00016672 | Sample type: blood |
| Strain, strain background (*Salmonella enterica* serovar Dublin) | ST10 strain used in phenotypic assays - this study. | Microbiological Diagnostic Unit, University of Melbourne | AUSMDU00016673 | Sample type: faeces |
| Strain, strain background (*Salmonella enterica* serovar Dublin) | ST10 strain used in phenotypic assays - this study. | Microbiological Diagnostic Unit, University of Melbourne | AUSMDU00016675 | Sample type: blood |
| Strain, strain background (*Salmonella enterica* serovar Dublin) | ST10 strain used in phenotypic assays - this study. | Microbiological Diagnostic Unit, University of Melbourne | AUSMDU00016677 | Sample type: blood |
| Strain, strain background (*Salmonella enterica* serovar Dublin) | ST10 strain used in phenotypic assays - this study. | Microbiological Diagnostic Unit, University of Melbourne | AUSMDU00016678 | Sample type: blood |
| Strain, strain background (*Salmonella enterica* serovar Dublin) | ST10 strain used in phenotypic assays - this study. | Microbiological Diagnostic Unit, University of Melbourne | AUSMDU00035676 | Sample type: blood |
| Strain, strain background (*Salmonella enterica* serovar Dublin) | ST10 strain used in phenotypic assays - this study. | Microbiological Diagnostic Unit, University of Melbourne | AUSMDU00056868 | Sample type: faeces |
| Strain, strain background (*Salmonella enterica* serovar Dublin) | ST4293 strain used in phenotypic assays - this study. | Microbiological Diagnostic Unit, University of Melbourne | AUSMDU00016679 | Sample type: urine |
| Strain, strain background (*Salmonella enterica* serovar Dublin) | ST74 strain used in phenotypic assays - this study. | Microbiological Diagnostic Unit, University of Melbourne | AUSMDU00014697 | Sample type: faeces |
| Strain, strain background (*Salmonella enterica* serovar Dublin) | ST74 strain used in phenotypic assays - this study. | Microbiological Diagnostic Unit, University of Melbourne | AUSMDU00016676 | Sample type: faeces |
| Strain, strain background (*Salmonella enterica* serovar Dublin) | ST74 strain used in phenotypic assays - this study. | Microbiological Diagnostic Unit, University of Melbourne | AUSMDU00016680 | Sample type: faeces |
| Strain, strain background (*Salmonella enterica* serovar Dublin) | ST74 strain used in phenotypic assays - this study. | Microbiological Diagnostic Unit, University of Melbourne | AUSMDU00016681 | Sample type: faeces |
| Strain, strain background (*Salmonella enterica* serovar Dublin) | ST74 strain used in phenotypic assays - this study. | Microbiological Diagnostic Unit, University of Melbourne | AUSMDU00016682 | Sample type: faeces |
| Strain, strain background (*Salmonella enterica* serovar Dublin) | ST74 strain used in phenotypic assays - this study. | Microbiological Diagnostic Unit, University of Melbourne | AUSMDU00021231 | Sample type: faeces |

*Appendix 1 Continued on next page*

*Appendix 1 Continued*

| Reagent type (species) or resource | Designation | Source or reference | Identifiers | Additional information |
|---|---|---|---|---|
| Strain, strain background (*Salmonella enterica* serovar Dublin) | ST1545 strain used in phenotypic assays - this study. | Microbiological Diagnostic Unit, University of Melbourne | AUSMDU00016683 | Sample type: faeces |
| Cell line (*Homo sapiens*) | THP-1, human monocyte | ATCC | ATCC TIB-202 | Cell line maintained in J. Pearson's lab. |
| Cell line (*Homo sapiens*) | HT-29, human colonic epithelial | ATCC | ATCC HTB-38 | Cell line maintained in J. Pearson's lab. |
| Antibody | anti-IL-1β (Asp 116), clone D3A3Z | Cell Signaling | Cat# 83186 | WB (1:1000) |
| Commercial assay or kit | CtyoTox 96 Non-radioactive Cytotoxicity Assay Kit | Promega | Cat#G1780 | |
| Commercial assay or kit | Pierce BCA Protein Assay Kit | Thermo Fisher Scientific | Cat#23225 | |
| Commercial assay or kit | DuoSet Human IL-1β/IL-1F2 ELISA | R&D Systems | Cat#DY201 | |
| Commercial assay or kit | Virus/Pathogen DSP midi kit | QIAGEN | Cat#937055 | |
| Commercial assay or kit | Ligation sequencing kit | Oxford Nanopore Technologies | Cat#SQK-LSK109 | |
| Software | Nullarbor pipeline | https://github.com/tseemann/nullarbor | v2.0.20191013 | |
| Software | Skesa | https://github.com/ncbi/SKESA | v2.4.0 skesa (RRID:SCR_024341) | |
| Software | mlst | https://github.com/tseemann/mlst | v2.19.0 MLST (RRID:SCR_010245) | |
| Software | Kraken | https://github.com/DerrickWood/kraken | v1.1.1 Kraken (RRID:SCR_005484) | |
| Software | Snippy | https://github.com/tseemann/snippy | v4.6.0 Snippy (RRID:SCR_023572) | |
| Software | *Salmonella* in silico Typing Resource (SISTR) | https://github.com/phac-nml/sistr_cmd | v1.1.1 sistr (RRID:SCR_024342) | |
| Software | Antimicrobial Resistance Identification by Assembly (ARIBA) | https://github.com/sanger-pathogens/ariba/tree/master | v2.14.5 Ariba (RRID:SCR_015976) | |
| Software | Prokka | https://github.com/tseemann/prokka | v1.14.6 Prokka (RRID:SCR_014732) | |
| Software | snippy-core | https://github.com/tseemann/snippy | v4.6.0 Snippy (RRID:SCR_023572) | |
| Software | Phaster | https://phaster.ca/ | v2020-12-22 | |
| Software | Gubbins | https://github.com/nickjcroucher/gubbins | v2.4.1 Gubbins (RRID:SCR_016131) | |
| Software | iqtree-calc_const_sites.sh | https://github.com/MDU-PHL/mdu-tools/blob/master/bin/iqtree-calc_const_sites.sh | IQ-TREE (RRID:SCR_017254) | |
| Software | IQ-TREE | http://www.iqtree.org/ | v2.1.0 IQ-TREE (RRID:SCR_017254) | |
| Software | *ggtree* | https://github.com/YuLab-SMU/ggtree | v2.3.3 ggtree (RRID:SCR_018560) | |
| Software | rhierBAPS | https://github.com/gtonkinhill/rhierbaps | v1.1.3 | |
| Software | AbriTAMR | https://github.com/MDU-PHL/abritamr | v0.2.2 | |
| Software | AMRFinder | https://github.com/ncbi/amr | v3.8.4 | |

*Appendix 1 Continued on next page*

*Appendix 1 Continued*

| Reagent type (species) or resource | Designation | Source or reference | Identifiers | Additional information |
|---|---|---|---|---|
| Software | ABRicate | https://github.com/tseemann/abricate | v1.0.1 Abricate (RRID:SCR_021093) | |
| Software | *igraph* | https://igraph.org/ | v1.2.4.1 igraph (RRID:SCR_019225) | |
| Software | invasive_*Salmonella* developed in 2018 | https://github.com/Gardner-BinfLab/invasive_salmonella | | |
| Software | BEAST | https://beast.community/ | v1.10.4 BEAST (RRID:SCR_010228) | |
| Software | TempEst | https://beast.community/ | v1.5.3 TempEst (RRID:SCR_017304) | |
| Software | BEAUti | https://beast.community/ | v1.10.4 BEAST2 (RRID:SCR_017307) | |
| Software | BEAGLE | https://beast.community/ | v3.0.2 BEAGLE (RRID:SCR_001789) | |
| Software | TreeAnnotator | https://beast.community/ | v1.10.4 BEAST2 (RRID:SCR_017307) | |
| Software | fastp | https://github.com/OpenGene/fastp | v0.20.1 fastp (RRID:SCR_016962) | |
| Software | Filtlong | https://github.com/rrwick/Filtlong | v0.2.1 Filtlong (RRID:SCR_024020) | |
| Software | Trycycler | https://github.com/rrwick/Trycycler | v0.4.1 | |
| Software | Medaka | https://github.com/nanoporetech/medaka | v1.4.3 | |
| Software | Unicycler | https://github.com/rrwick/Unicycler | v0.4.4 Unicycler (RRID:SCR_024380) | |
| Software | Bandage | https://github.com/rrwick/Bandage | v0.8.1 Bandage (RRID:SCR_022772) | |
| Software | Prokka | https://github.com/tseemann/prokka | v1.14.6 Prokka (RRID:SCR_014732) | |
| Software | Bakta | https://github.com/oschwengers/bakta | v1.10.1 Bakta (RRID:SCR_026400) | |
| Software | Blast Ring Image Generator (BRIG) | https://github.com/happykhan/BRIG | v0.95 BRIG (RRID:SCR_007802) | |
| Software | mob-typer | https://github.com/phac-nml/mob-suite | v3.1.0 | |
| Software | Easyfig | https://mjsull.github.io/Easyfig/ | v2.2.2 Easyfig (RRID:SCR_013169) | |
| Software | gggenomes | https://github.com/thackl/gggenomes | v0.9.12.9000 | |
| Software | Prism software | GraphPad Software | v9.0 GraphPad (RRID:SCR_000306) | |
| Software | Panaroo | https://github.com/gtonkinhill/panaroo | v1.3.4 Panaroo (RRID:SCR_021090) | |
| Software | snp-dists | https://github.com/tseemann/snp-dists | v0.8.2 | |

