## [Editor Report · eLife Assessment]

This study provides the most comprehensive analysis of *Salmonella* Dublin to date, uncovering distinct genotypic adaptations, antimicrobial resistance patterns, and virulence strategies that influence epidemiological success. The revised manuscript is very **valuable**, rigorous and **compelling**.

---

## [Referee Report · Reviewer #1 (Public review)]

The manuscript consists of two separate but interlinked investigations: genomic epidemiology and virulence assessment of *Salmonella* Dublin. ST10 dominates the epidemiological landscape of S. Dublin, while ST74 was uncommonly isolated. Detailed genomic epidemiology of ST10 unfolded the evolutionary history of this common genotype, highlighting clonal expansions linked to each distinct geography. Notably, North American ST10 was associated with more antimicrobial resistance compared to others. The authors also performed long read sequencing on a subset of isolates (ST10 and ST74), and uncovered a novel recombinant virulence plasmid in ST10 (IncX1/IncFII/IncN). Separately, the authors performed cell invasion and cytotoxicity assays on the two S. Dublin genotypes, showing differential responses between the two STs. ST74 replicates better intracellularly in macrophage compared to ST10, but both STs induced comparable cytotoxicity levels. Comparative genomic analyses between the two genotypes showed certain genetic content unique to each genotype, but no further analyses were conducted to investigate which genetic factors likely associated with the observed differences. The study provides a comprehensive and novel understanding on the evolution and adaptation of two S. Dublin genotypes, which can inform public health measures. The methodology included in both approaches were sound and written in sufficient detail, and data analysis were performed with rigour. Source data were fully presented and accessible to readers.

Comments on revised version:

The authors have addressed all the points raised by the reviewer. The manuscript is now much enhanced in clarity and accuracy. The rewritten Discussion is more relevant and brings in comparison with other invasive *Salmonella* serotypes.

---

## [Referee Report · Reviewer #2 (Public review)]

This is a comprehensive analysis of *Salmonella* Dublin genomes that offers insights into the global spread of this pathogen and region-specific traits that are important to understand its evolution. The phenotyping of isolates of ST10 and ST74 also offer insights into the variability that can be seen in S. Dublin, which is also seen in other Salmonella serovars, and reminds the field that it is important to look beyond lab-adapted strains to truly understand these pathogens. This is a valuable contribution to the field. The only limitation, which the authors also acknowledge, is the bias towards S. Dublin genomes from high-income settings. However, there is no selection bias; this is simply a consequence of publicly available sequences.

---

## [Author Response]

The following is the authors’ response to the previous reviews

**Public Reviews:**

**Reviewer #1 (Public review):**
The manuscript consists of two separate but interlinked investigations: genomic epidemiology and virulence assessment of *Salmonella* Dublin. ST10 dominates the epidemiological landscape of S. Dublin, while ST74 was uncommonly isolated. Detailed genomic epidemiology of ST10 unfolded the evolutionary history of this common genotype, highlighting clonal expansions linked to each distinct geography. Notably, North American ST10 was associated with more antimicrobial resistance compared to others. The authors also performed long read sequencing on a subset of isolates (ST10 and ST74), and uncovered a novel recombinant virulence plasmid in ST10 (IncX1/IncFII/IncN). Separately, the authors performed cell invasion and cytotoxicity assays on the two S. Dublin genotypes, showing differential responses between the two STs. ST74 replicates better intracellularly in macrophage compared to ST10, but both STs induced comparable cytotoxicity levels. Comparative genomic analyses between the two genotypes showed certain genetic content unique to each genotype, but no further analyses were conducted to investigate which genetic factors likely associated with the observed differences. The study provides a comprehensive and novel understanding on the evolution and adaptation of two S. Dublin genotypes, which can inform public health measures. The methodology included in both approaches were sound and written in sufficient detail, and data analysis were performed with rigour. Source data were fully presented and accessible to readers.Comments on revised version:The authors have addressed all the points raised by the reviewer. The manuscript is now much enhanced in clarity and accuracy. The re-written Discussion is more relevant and brings in comparison with other invasive *Salmonella* serotypes.Comments:In light of the metadata supplied in this revision, for Australian isolates, all human cases of ST74 (n=7) were from faeces (assuming from gastroenteritis) while 18/40 of ST10 were from invasive specimen (blood and abscess). This may contradict with the manuscript's finding and discussion on different experiment phenotypes of the two STs, with ST74 showing more replication in macrophages and potentially more invasive. Thus, the reviewer suggests the authors to mention this disparity in the Discussion, and discuss possible reasons underlying this disparity. This can strengthen the author's rationale for further in vivo studies.

We thank the reviewer for pointing out this important observation. We have amended the text in the Discussion to address the differences in source of human cases as suggested by the Reviewer (lines 392-430). We have also included text highlighting the important knowledge gaps in understanding the drivers for emerging iNTS with broad host ranges and identify future avenues of research that could be explored to better understand the observed differences in the host-pathogen interactions.

**Reviewer #2 (Public review):**
This is a comprehensive analysis of *Salmonella* Dublin genomes that offers insights into the global spread of this pathogen and region-specific traits that are important to understand its evolution. The phenotyping of isolates of ST10 and ST74 also offer insights into the variability that can be seen in S. Dublin, which is also seen in other Salmonella serovars, and reminds the field that it is important to look beyond lab-adapted strains to truly understand these pathogens. This is a valuable contribution to the field. The only limitation, which the authors also acknowledge, is the bias towards S. Dublin genomes from high income settings. However, there is no selection bias; this is simply a consequence of publicly available sequences.

We thank the reviewer for their comments and acknowledge the limitations of this study.